# Tonal Behavior as of Areal and Typological Concerns: Centering on the Sinitic and Kam-Tai Languages in Lingnan

Hanbo Liao 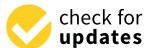

Department of Linguistics, The University of Hong Kong, Pok Fu Lam, Hong Kong SAR, China; mayxreux@gmail.com

**Abstract:** From the perspective of areal linguistics, this paper examines the similarities in tonal behavior between Sinitic and Kam-Tai, the two most populous language groups in Lingnan. By relying on some frameworks for investigating tone systems, i.e., tone-box theories, which largely involve the evolution of tones, the following duplicating patterns and paths of diffusion of areal features are identified. (1) The secondary tonal split conditioned by vowel length on checked syllables, as well as the secondary tonal split of the upper-register tones conditioned by the laryngeal features of initial consonants, both originated in Kam-Tai languages and have diffused into some neighboring Sinitic languages. (2) The pattern of the secondary tonal split of the lower-register tones conditioned by laryngeal features of the initial consonants originated in northern authoritative Sinitic languages and spread widely among different subgroups of Sinitic languages; its diffusion into the Kam-Tai languages is limited to the lexical category of loanwords. (3) The upper-register tones associated with sonorant initials found in Lingnan Sinitic languages are suggested to be of a Kam-Tai origin trait. Further, their underlying areal typological rules are also summarized, concentrating on different upper limits for the possible secondary tonal split in the Sinitic and Kam-Tai languages, which were determined by the historically distinct laryngeal features of the initial consonants of the two language groups.

**Keywords:** Lingnan; Sinitic; Kam-Tai; areal linguistics; tone-box; language typology; tonology

## 1. Introduction: Tones as an Areal Feature Diffusing in Sinitic and Kra-Dai

This paper is a study on the tonal systems of the languages spoken in the Lingnan region, centered on Guangxi and Guangdong, China. Among the languages of the world, Sinitic and Kam-Tai are perhaps two of those language groups with the most complex tonal split patterns in the history of tonal development, both from diachronic and synchronic points of view. The core Lingnan region[1] roughly equals the western Lingnan sprachbund proposed by Szeto and Yurayong (2022) and is the geographical area in which the target languages of this paper, Sinitic and Kam-Tai, have the most concentrated contact. In this paper, I present various tonal split patterns and tonal behavior of the Lingnan languages in these two language groups, determine areal features in tonal split patterns emerging as a result of intense language contact and extensive diffusion, take into account the diachronic depth and the synchronic breadth, and ultimately conclude with the number of tonological areal types.

To present, at a glance, the position of Sinitic and Kam-Tai languages in their respective language families, we present tree diagrams of their affiliation in Figures 1 and 2.

According to the subdivisions and affinities within the two linguistic groups illustrated in the two figures above, there are two major points that need to be noted as premises for the discussion in this paper. The first is that the non-Mandarin Sinitic languages, including Cantonese, Pinghua, and Hakka in Lingnan, all have colloquial readings of Chinese etyma inherited from older phonological layers, as well as literary readings adapted from the authoritative northern dialects of the Late Middle Chinese (LMC) periods. The

second point is that the Kra-Dai languages discussed in this paper, including the Zhuang and Kam languages of core Lingnan, as well as Thai and other Southwestern Tai languages on the MSEA, are all of the most populous branch of Kam-Tai, particularly the Kam-Sui and Tai subbranches, which have had intense and long-standing contact with Lingnan Sinitic languages in history. The non-Kam-Tai languages of the peripheral Lingnan, namely Kra and Hlai, as well as Be of Tai-Be, are not included in the main scope.

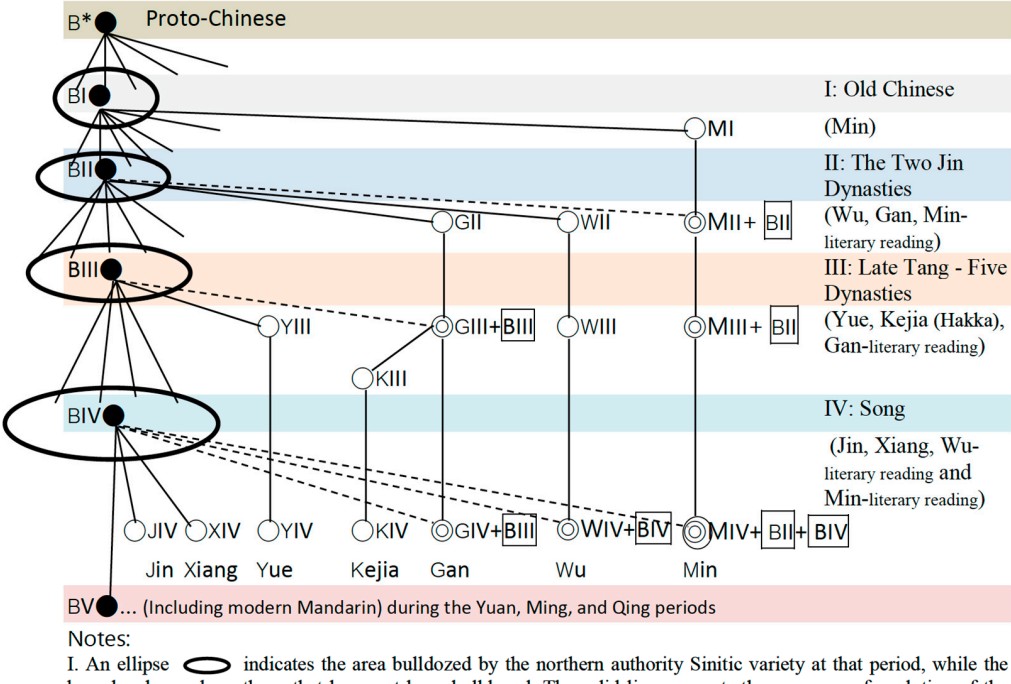

**Figure 1.** H. Wang's (2009, pp. 212–14) model of Sinitic subgrouping and classification.

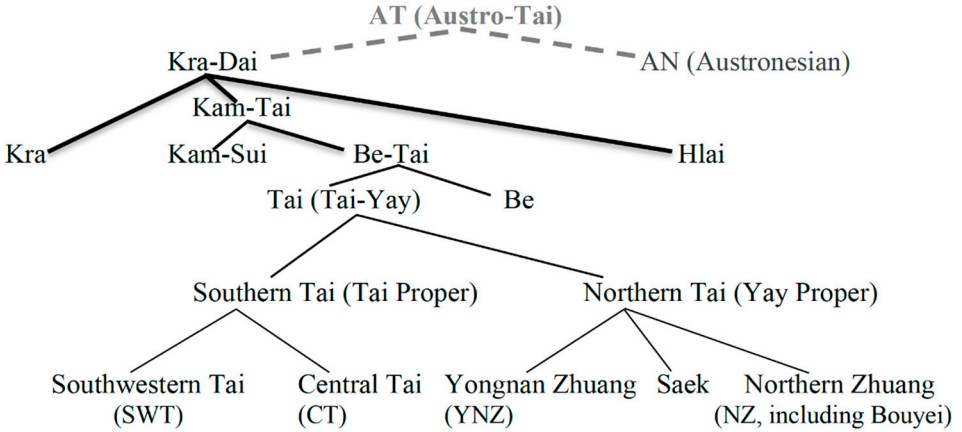

**Figure 2.** A Kra-Dai language family tree: Austro-Tai hypothesis (Ostapirat 2005), Kam-Tai division, and two plus three Tai taxonomy (Liao and Tai 2019).

An important phonological trait for Lingnan as part of Mainland Southeast Asia (MSEA) is the similar tone systems among numerous languages, despite them coming from different language families, namely Sinitic, Kra-Dai, Hmong-Mien, and Việt–Mường. This similarity is first demonstrated by the fact that the modern tone systems in these languages all reflect a proto-system of "three-plus-one", i.e., three tonal categories *A, *B, and *C on "smooth syllables", syllables with continuant (vowel/sonorant) endings, plus a tonal category *D on a non-tonal checked syllable, and syllables with oral stop endings (such as -p, -t, and -k, mostly unreleased)[2]. It is further illustrated by the neat correspondence of the later 8–10 tonal categories as the result of register tonal splits among the above languages. Such neat correspondence of tone systems among these languages has been demonstrated in previous studies, such as Haudricourt's (1954) tonal correspondences between Vietnamese and Chinese, and Downer's (1963) tonal correspondences between Chinese, Thai, and Hmong-Mien.

To clarify how the neat tonal correspondences among these languages formed, it is first necessary to determine whether tones in these languages are innate or an acquired characteristic. One of the methodological aspects of this paper is indeed based on the theories of the origin and development of tones. Previous studies have differed regarding whether tone is an inherited feature or an areal trait in a tonal language. Some early studies, such as Maspero (1912), suggested that tone was an inherent feature of tonal languages, i.e., that these languages had tones from the beginning. However, since Haudricourt (1954) suggested that Vietnamese tones arose from the decline of specific consonantal rime endings, other scholars have proposed a similar process for the emergence of tones in Sinitic, Hmong-Mien, and Kra-Dai languages, which Matisoff (1973) refers to as tonogenesis. Since then, most historical and comparative linguists have tended to attribute the tones of these languages to a later development, and to an areal trait that began in Chinese and diffused to other languages, as demonstrated in Section 2. On this premise, some common tonal behaviors represented by abundant secondary tonal split patterns in Lingnan Sinitic and Kam-Tai languages, the main objects of this study, can hardly be regarded as inherited in these languages, as they developed on the basis of earlier tones, which have already been suggested to be later developments themselves. Thus, exploring the origins and the direction of the diffusion of some common tonal behaviors becomes one of the key findings of this paper. Another methodological aspect is to improve the inadequate use of tone-box frameworks, which deal with tonal split theories, including primary and secondary splits, according to the laryngeal features of initial consonants at the time of the tonal split, for indicating the currently complicated tonal correspondence among sister languages. These methods will be maximized in this paper to reveal new insights and cross-family correlations from the empirical data that have significantly increased in recent decades. Put simply, by using these updated and improved tone-box frameworks to present the complicated tonal correspondences at a glance, we can easily identify tonal behaviors that were previously not readily apparent as cross-family. If a tonal behavior is widespread among all the subbranches of the Kam-Tai languages, but has limited presence in the neighboring Sinitic languages and is not found in Sinitic languages elsewhere, then we can rule out its Sinitic origin, and vice versa.

Based on a combination of previous research findings and empirical tonal data from cross-family correlations, this paper finds that different tonal split patterns and tonal behaviors may have different origins and may have diffused in opposite directions. Previous studies have pointed to secondary tonal split patterns conditioned by vowel length in checked syllables, which originated in Kam-Tai and diffused to some Lingnan Sinitic languages. However, although it is also found that a secondary split of upper-register tones conditioned by laryngeal features of originally voiceless initial consonants is found in both Kam-Tai and some Lingnan Sinitic languages, little has been done to associate them together. In this paper, a cross-family comparison of this topic reveals that this trait also originated in Kam-Tai and diffused into some Lingnan Sinitic languages, particularly the Cantonese and Pinghua dialects. In contrast, for the secondary split of lower-register tones

conditioned by laryngeal features of originally voiced initial consonants, similar cross-family comparisons reveal that the direction of origin and diffusion is reversed. A hypothesis of the lexical diglossia of colloquial vs. literary readings in Lingnan Sinitic languages, particularly the *zhuóshǎng guīqù* phenomenon in tone B2, is given to explain the formation of this trait, together with an approach to deal with this phenomenon as "borrowings" in both the Lingnan Sinitic and Kam-Tai languages in a conventional sense.

Apart from secondary tonal split patterns, there is another notable tonal behavior presenting as a number of lexicons with sonorant initials associated with the upper-register tones in Lingnan Sinitic languages, such as Cantonese and Pinghua. These Lingnan Sinitic languages are generally considered to be direct descendants of Middle Chinese (MC), which did not have voiceless sonorant initial consonants; thus, such tonal behavior is anomalous. A cross-family comparison reveals new insights that a large proportion of these lexical items in Lingnan Sinitic languages are likely to be of Kam-Tai origin, either substratum or loanwords.

Furthermore, the tonal evidence in this paper can be used to propose a more fine-grained relative chronology of the changes under contact between Sinitic and Kra-Dai, as well as criteria to detect Kam-Tai borrowings in southern Sinitic. A typical example is that the number of Sinitic loanwords with the *zhuóshǎng guīqù* phenomenon shared among Tai dialects from all the terminal Tai subgroups suggests that the *zhuóshǎng guīqù* phenomenon occurred before the latest limit of the primary tonal split, i.e., before SWT split from CT languages, as the register tonal split in Kam-Tai languages occurred much later than in Chinese because it was a trait adapted from LMC, and Thai did not have a primary tonal split until around 700 years ago (as demonstrated in Sections 2.1 and 3.2).

In short, the above key points are those highlighted as the methodological contribution of this paper to the field, aiming at filling research gaps by exploring secondary tonal split patterns and tonal behavior as areal features and some other specific tonal behaviors from an areal linguistic perspective. In the following discussion, we will introduce the theoretical frameworks of this study in the two subsections of Section 2, with a large amount of specific detail on the phenomenon and related theoretical matters; Section 3 and its subsections will introduce the major findings of this study, revealing the origins and directions of the diffusion of secondary tonal split patterns and some specific areal tonal behavior as a result of language contact; and Section 4 concludes this paper.

## 2. Theoretical Basis: Tonology Applied to Sinitic and Kam-Tai languages

### 2.1. Tonogenesis and Tonal Splits

We have mentioned the similarities of tone systems among the language families of Sinitic, Kra-Dai, Hmong-Mien, and Việt–Mường, and have pointed out that these similarities are not the result of the innate or separate development of the languages mentioned above, but rather the result of the diffusion of tone traits. In this diffusion, loanwords borrowed from Chinese, including Old Chinese (OC), MC, and other historical predecessors of the Chinese language, played a key role. The process of borrowing began with the emergence of tones. Tones emerged in Chinese some 1500 years ago (cf. Pulleyblank 1978), while research into other languages also presents similar theories about the emergence and development of tones. For the first step, tonogenesis, these languages developed from non-tonal languages due to the loss or degradation of two rhyme endings, *-ʔ and *-h (← *-s), because syllables with these two endings were semantically discriminative to plain syllables without these two endings and those ending with oral stops (*-p/-t/-k). As a result, three early tones, *A, *B, and *C, on smooth syllables were developed to maintain semantic distinctions, while syllables ending in *-p/-t/-k were designated as "checked syllables" and the pitch on them was tone *D, presenting an *ABC/*D or 3/1 tone category pattern (cf. Handel 2014; Hill 2019). The generalization and definition of the tonogenesis of languages in MSEA can be seen in Matisoff (1973), while the respective tonogenesis of different language families/groups can be seen in some of the following studies: Haudricourt (1954) for Vietnamese, Pulleyblank (1962), Mei (1970, p. 86) and Norman (1988, pp. 55–57) for

Chinese, Chang (1972) for Hmong-Mien, Pittayaporn (2009, pp. 270–71) and Liao (2016a, pp. 81–84; 2016b, pp. 96–120) for Tai, and Sagart (2019) and Liao (2020) for Kra-Dai and Tai. However, the triggers for tonogenesis may vary, with mainstream studies suggesting that, for non-Sinitic groups, the triggers may be contact-induced, specifically by contact with Chinese, which had a strong historical influence.

Pittayaporn (2014) discussed diachronic layers of Chinese loanwords in the Tai languages that spanned more than 1000 years from OC to Late Middle Chinese (LMC). Although his intention was to verify the timing of the migration of the southwestern Tai language group from its ancestral homeland of Guangxi–North Vietnam into the core areas of MSEA, we can also see the temporal depth and breadth of Chinese loanwords in the Tai languages. For example, when borrowing a term with the sibilant final *-s from non-tonal OC (Baxter and Sagart 2014b), such as *s.li[j]-s 'four' (cf. Baxter and Sagart 2014a, p. 104), the non-tonal, early proto-Tai might have used its aspirated final *-h to adapt this term, namely *ɬi:h 'four' (Liao 2020). Later, as OC *-s became the MC *-h, which gave rise to a tone, tone *C/tone qù (去聲), to preserve semantic distinctions, the term 'four' changed to *sij$^C$ in Early Middle Chinese (EMC) (via *sijH) (cf. Baxter and Sagart 2014a, p. 104). Under this influence, the *-h of proto-Tai was also dropped off to develop a tone B, so that, in the tonal late proto-Tai (Liao 2016b, p. 94), *ɬi:h 'four' became *ɬi:$^B$. Thus, a correspondence between the Chinese tone *C/qù and Tai tone *B was formed. Such tonal correspondence among the early tone systems of these different language groups, and their historical sources that are commonly suggested are summarized in Table 1.

**Table 1.** Correspondence of the early tone systems among Chinese, Kra-Dai, Vietnamese, and Hmong-Mien.

| Suggested Origin | EMC | Vietnamese | Kra-Dai | Hmong-Mien |
|---|---|---|---|---|
| - | A (*píng*/level) | *A (*ngang-huyền*) | *A | *A |
| *-ʔ | B (*shǎng*/rising) | *B (*sắc-nặng*) | *C | *B |
| *-s > *-h/*-h | C (*qù*/departing) | *C (*hỏi-ngã*) | *B | *C |
| *-p/t/k | D (*rù*/entering) | *D (*sắc-nặng*) | *D | *D |

For the second step, further tonal splits from these four early tone categories in each of the above groups were also very similar in the split patterns. Unlike tonogenesis, which began with the loss of final syllable consonants, tonal splits are conditioned by the loss of voicing contrast in initial consonants. As a major areal trait in both China and MSEA, it is suggested to have begun in LMC around 1000 years ago, during the late Tang period, when the voicing contrast of initial consonants began to be lost, mainly through the devoicing of the original voiced initial stops to cause the merge into voiceless initial stops (Norman 1988, p. 53). Such a loss of voicing contrast in initials led to a further split of the early tones into two pitch registers, the high (upper) and low (lower) tones conditioned by originally voiceless and voiced initials, respectively. As a result, the original four tones became eight tones, i.e., *ABC/*D > A1, A2, B1, B2, C1, C2/D1, and D2, or 3/1 > 6/2 (in which tones before the slash are on smooth syllables and those behind the slash are on checked syllables), and this pattern was clearly replicated in Vietnamese and Tai (Norman 1988, pp. 54–55)[3]. We know from previous studies, such as Downer (1963) and Ballard (1981), that tonal splits of the Hmong-Mien languages adopted this pattern without exception. It is speculated that the devoicing of early voiced initial stops and register tonal split, two traits of the MSEA, began to spread around 1000 years ago from Chinese to all the above languages. This pattern was, however, merely the first step in the register tonal split for Kam-Tai languages, so it is called a primary tonal split (Liang and Zhang 1993, 1996, p. 62; Zhang 1980; Zhang et al. 1999, p. 243; Liao 2016a, pp. 20–22; 2016b, pp. 124–26). Using Table 2 as an example, the neat correspondence between register tonal split patterns in LMC and Kam-Tai is

presented. In particular, the traditional Chinese phonetic terms (e.g., *píng*/level) refer to Chinese tones, while ABCD refers to Kam-Tai tones in the primary tonal split pattern.

**Table 2.** Sinitic tonal split pattern and the identical Kam-Tai primary tonal split pattern.

| MC/PKT Tone \\ MC/PKT Initial | Píng/*A (平 Level) | Qù/*B (去 Departing) | Shǎng/*C (上 Rising) | Rù/*D (入 Entering) |
|---|---|---|---|---|
| Voiceless | Yīnpíng/A1 (陰平 Upper level) | Yīnqù/B1 (陰去 Upper departing) | Yīnshǎng/C1 (陰上 Upper rising) | Yīnrù 陰入/D1 (陰入 Upper entering) |
| Voiced | Yángpíng/A2 (陽平 Lower level) | Yángqù/B2 (陽去 Lower departing) | Yángshǎng/C2 (陽上 Upper rising) | Yángrù 陽入/D2 (陽入 Upper entering) |

Again, in the formation of such a neat correspondence between register tonal split patterns, it was Chinese loanwords that played a decisive role in this process. Previous studies, such as Wulff (1934), Manomaivibool (1975), and Li (1976), established the correspondence between Tai and Chinese in terms of the vocabulary shared between them. Most of the words that make up the shared vocabulary appear to be MC loanwords borrowed into Tai. This can be demonstrated in the tonal correspondence between Cantonese and Debao Urban Yang Zhuang (a Central Tai language) that Liao (2016b, p. 97) compiled, as revised in Table 3, with additional data from the Yang Zhuang variety of Jingxi Urban. The mutually corresponding tonal categories from the three language varieties may have different tonal values, but maintain a neat correspondence (as seen between the gloss series 1 and 2). This indicates the systematic nature of historical Chinese loanwords in the Tai languages.

**Table 3.** Tonal correspondence between Cantonese and two Yang Zhuang varieties.

| Tonal Category | A1 | A2 | B1 | B2 | C1 | C2 | DS1 | DL1 | DS2 | DL2 |
|---|---|---|---|---|---|---|---|---|---|---|
| **Tai Tone in the Sinitic Circles** | 1 | 2 | 5 | 6 | 3 | 4 | 7 | 9 | 8 | 10 |
| Chinese terms | yīn píng | yáng píng | yīn qù | yáng qù | yīn shǎng | yáng shǎng | yīn rù (short) | yīn rù (long) | yáng rù (short) | yáng rù (long) |
| Tonal value — Cantonese | 55 | 21/11 | 33 | 22 | 35 | 24 | 55 | 33 | 22 | 22 |
| Tonal value — DB Yang Z. | 453 | 31 | 55 | 33 | 24ʔ | 213ʔ | 45 | 55 | 33 | 33 |
| Tonal value — JX Yang Z. | 53 | 31 | 45 | 132 | 33ʔ | 213ʔ | 33 | 45 | 21 | 13 |
| Gloss Series 1 | 開 open | 時 time | 過 to pass | 伴 company | 廣 wide | 馬 horse | 七 seven | 發 deliver | 佛 Buddha | 抹 wipe |
| Cantonese | hɔi⁵⁵ | si²¹ | kwɔ³³ | pun²² | kwɔŋ³⁵ | ma²⁴ | tsʰɐt⁵⁵ | fat³³ | fɐt²² | mat²² |
| DB Yang Z. | kʰɐj⁴⁵³ | ɬəj³¹ | kwa⁵⁵ | pu:n³³ | ku:ŋ²⁴ʔ | ma²¹³ʔ | tɕɐt⁴⁵ | fa:t⁵⁵ | pɐt³³ | ma:t³³ |
| JX Yang Z. | kʰɐj⁵³ | ɬəj³¹ | kwa⁴⁵ | pu:n¹³² | ku:ŋ³³ʔ | ma²¹³ʔ | tɕɐt³³ | fa:t⁴⁵ | pɐt²¹ | ma:t¹³ |
| Gloss Series 2 | 書 book | 頭 first | 救 save | 用 use | 炒 to fry | 老 senior | 北 north | 八 eight | 十 ten/October | 蠟 candle |
| Cantonese | sy⁵⁵ | tʰɐw²¹ | kɐw³³ | jʊŋ²² | tʃʰau³⁵ | low²⁴ | pɐk⁵⁵ | pat³³ | sɐp²² | lap²² |
| DB Yang Z. | ɬøɥ⁴⁵³ | tɐw³¹ | kjɐw⁵⁵ | jʊŋ³³ | ɕa:w²⁴ʔ | la:w²¹³ʔ | pɐk⁴⁵ | pe:t⁵⁵ | tɕɐp³³ | la:p³³ |
| JX Yang Z. | ɬəj⁵³ | tɐw³¹ | kjɐw⁴⁵ | jʊŋ¹³² | ɕa:w³³ʔ | la:w²¹³ʔ | pɐk³³ | pe:t⁴⁵ | tɕɐp²¹ | la:p¹³ |

Note that the glottal symbol -ʔ following the tonal values 24ʔ, 33ʔ or 231ʔ in Table 3 refers to glottalization of these tones. It is a remnant of the historical glottal stop rime ending in these Tai dialects after tone C had been created by the weakening of that glottal stop ending. From Table 3, we can see that, unlike the tonal correspondence presented in Table 2, the D tones further split in addition to the upper and lower registers due to the different vocalic lengths of the syllable rimes to form a pattern of four tonal categories.

Among them, Cantonese and Debao Urban Yang Zhuang have this secondary split in the upper register only, so in the checked syllables, there are, in fact, three tones. Note that these checked syllable tones are often analyzed as allotones of tones on smooth syllables with the same or similar pitch contour in modern phonology because of the complementary distribution between them. Such a condition of the secondary tonal split, namely the vowel length on the checked syllables conditioning further possible tonal splits, is commonly found in Kra-Dai dialects, particularly the Kam-Sui and Tai subbranches, including Southern Kam, Sui, Thai, Lao, and many Zhuang varieties. Thus, in Kam-Tai languages, the tone category patterns can be summarized as an integrated pattern of A1, A2, B1, B2, C1, C2, DL1, DL2, DS1, and DS2, and because of some modern tonal mergers on smooth syllables, the number of tones may change into some patterns from 5/3 to 6/3 or 6/4, in which the tone numbers before the slash are on smooth syllables and those behind the slash are on checked syllables. This trait has diffused into the Sinitic languages in the core Lingnan area, Yue, and Pinghua, such as the 6/3 pattern in Cantonese (Guangzhou and Hong Kong) shown in Table 3, which is not found in non-Lingnan Sinitic languages (de Sousa 2015a, pp. 168–69; 2015b, pp. 368–71).

　　In summary, tonal development is involved in diachrony. As an example, Figure 3 shows a full diachronic picture of the tonal development in the Sinitic languages, including those in Lingnan and non-Lingnan areas (except Min Chinese). In particular, Cantonese represents the secondary patterns developed on the basis of the eight tones of LMC, while Standard Mandarin represents those merging some of the tones of the eight-tone pattern.

Stage 1: Old Chinese (first millennium BC): no tones but four types of phonetic syllable contrast

| voiceless initial | vowels ending in sonorants | vowels-ʔ | vowels-s | vowels-p/-t/-k |
| --- | --- | --- | --- | --- |
| voiced initial | | | | |

⇩　　　　⇩　　　　⇩　　　　⇩

Stage 2: Early Middle Chinese (around 6th Century AD): phonemic four tones

| voiceless initial | *píng* | *shǎng* | *qù* | *rù* |
| --- | --- | --- | --- | --- |
| voiced initial | | | | |

⇩　　　　⇩　　　　⇩　　　　⇩

Stage 3: Tonal splits to phonemic eight tones in Late Middle Chinese (around 1000 year BP.)

| Initial voicing | *yīn píng* | *yīn shǎng* | *yīn qù* | *yīn rù* |
| --- | --- | --- | --- | --- |
| mergers | *yáng píng* | *yáng shǎng* | *yáng qù* | *yáng rù* |

⇩　　　　⇩　　　　⇩　　　　⇩

Stage 4: Modern Chinese: tones preserved, or merged, or split from 8-tone system.

e.g. *Guǎngzhōu* (Cantonese/Standard Yue)

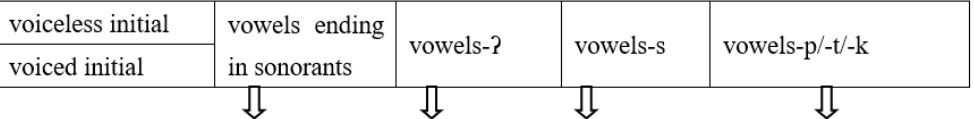

| Preservation & split of LMC tones | *yīn píng* | *yīn shǎng* | *yīn qù* | split > | short *yīn rù* |
| --- | --- | --- | --- | --- | --- |
| | | | | | long *yīn rù* |
| | *yáng píng* | *yáng shǎng* | *yáng qù* | *yáng rù* | |

e.g. *Běijīng* (Standard Mandarin):

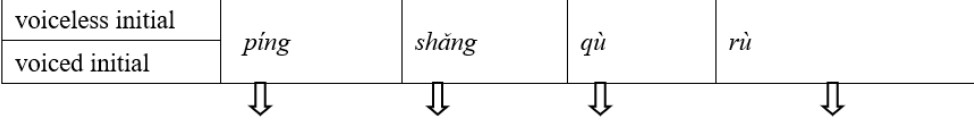

| Preservation & mergers of LMC tones | *yīn píng* | Merger > *shǎng* | Merger >*qù* | Merger > *one of the other four tones* |
| --- | --- | --- | --- | --- |
| | *yáng píng* | | | |

**Figure 3.** Chinese tonal development scheme (cf. Liao 2016b, p. 62).

Although it is commonly agreed that the traits of tonogenesis and tonal register splits occurred first in Chinese and affected other languages in South China and MSEA as a whole, as the further secondary tonal split pattern on checked syllables demonstrates above, certain specific details of tonal development have taken on a character of their own in MSEA (including Kam-Tai and Sinitic in Lingnan), that is, they have given rise to MSEA-specific traits that are distinct from those of the common Sinitic languages. However, previous studies on languages in Lingnan did not sort out the secondary tonal split patterns beyond those on checked syllables and did not treat them as a trait. The result is that these phenomena are only seen in individual languages, ignoring the fact that other languages in the region also share these features, obscuring the essence of secondary tonal split patterns as a significant areal trait.

*2.2. Tonal Split Patterns Displayed in a Tone-Box Framework*

As a tool and theoretical framework for investigating secondary tonal splits, tone-box charts designed to address the secondary tonal split patterns of the Kam-Tai languages form an important basis. This is because the tonal splits of the Kam-Sui and Tai branches of Kam-Tai are by far the most varied and complex of any of the language groups, although the tonal split is an areal phenomenon, as demonstrated above, found in the Sinitic, Kra-Dai, Hmong-Mien, and Vietic languages. A review of the Kam-Tai tone-box theories, which are commonly used, is called for when we need to take a comprehensive look at secondary tonal split patterns and provide a similar theoretical framework for the Sinitic languages of Lingnan.

We have illustrated the neat correspondence between the tone system of Kam-Tai and that of Chinese (especially in Lingnan Sinitic languages, such as Cantonese and Pinghua) by introducing the role played by Chinese loanwords. In practice, however, we find that many words still have seemingly irregular correspondences due to the laryngeal features of initial consonants. This is the result of what we call secondary tonal splits in Kam-Tai. This is the most significant difference between Kam-Tai and Sinitic languages in the tonal split patterns. Among the tonal split patterns of Kam-Tai languages, in addition to historically occurring with the primary tonal splits (similar to Sinitic as in Table 2), in nearly half of the dialects, there are also secondary tonal splits, i.e., in the upper-register tones, where different laryngeal features of initial consonants (such as aspiration and glottalization) may later condition further splits. These secondary tonal splits can be shown at a glance by using a tone-box chart. Among various tone-box frameworks, Gedney's ([1972] 1989) tone-box chart has been commonly used in the study of the tonal systems of the Southwestern Tai dialects, as seen in its application to the tones of Standard Thai and Northern Thai (Chiang Rai) shown in Tables 4 and 5, respectively.

Unlike the Sinitic two series of initials, voiceless and voiceless, in order to accurately describe the upper-register secondary tonal splits that are often found in the Southwestern Tai dialects, Gedney's formulation subdivides the original upper series (voiceless) into three sets based on different laryngeal features, namely voiceless friction (including voiceless aspirated stops and voiceless continuants) and voiceless unaspirated stops, and glottal (glottal stop and preglottalized stops/sonorants)[4], which conditioned possible tonal splits in any Southwestern Tai dialect. Note that the original lower series remains unchanged (as no secondary splits on lower-register tones are found in Tai dialects). As shown in Tables 4 and 5, boxes with the same color indicate those with the same tone, and checkered syllable tones being regarded as allotones of smooth syllable tones, although their tone values may have slightly different contours from the toneme on smooth syllables due to the differences in syllable structure.

**Table 4.** Gedney's tone-box chart (Gedney [1972] 1989, p. 202) applied to Standard Thai.

| Initials at Time of Tonal Splits | Proto-Tai Tones | | | | |
|---|---|---|---|---|---|
| | **A** | **B** | **C** | **DL** | **DS** |
| Voiceless friction sounds, *s-, *hm-, *ph-, etc. | $hu{:}^{24}$ 'ear'; $k^ha{:}^{24}$ 'leg' | $k^haj^{11}$ 'egg'; $p^ha{:}^{11}$ 'to split' | $k^ha{:}^{41ʔ}$ 'to kill'; $sia^{41ʔ}$ 'shirt' | $k^ha{:}t^{11}$ 'torn'; $ŋiak^{11}$ 'gums' | $mat^{11}$ 'flea'; $p^hak^{11}$ 'vegetable' |
| Voiceless unaspirated stops, *p-, *t-, *k-, etc. | $pi{:}^{33}$ 'year'; $ta{:}^{33}$ 'eye' | $pa{:}^{11}$ 'forest'; $kaj^{11}$ 'chicken' | $ka{:}w^{41ʔ}$ 'nine'; $tom^{41ʔ}$ 'to boil' | $pɔ{:}t^{11}$ 'lung'; $tɔ{:}k^{11}$ 'to pound' | $kop^{11}$ 'frog'; $tap^{11}$ 'liver' |
| Glottal, *ʔ-, *ʔb-, etc. | $bin^{33}$ 'to fly'; $dɛ{:}ŋ^{33}$ 'red' | $da{:}^{11}$ 'to scold'; $ba{:}^{11}$ 'shoulder' | $ba{:}^{41ʔ}$ 'crazy'; $ba{:}n^{41ʔ}$ 'village' | $dɛ{:}t^{11}$ 'sunshine'; $ʔa{:}p^{11}$ 'to bathe' | $bet^{11}$ 'fishhook'; $ʔok^{11}$ 'chest' |
| Voiced, *b-, *m-, *l-, *z-, etc. | $mɨ{:}^{33}$ 'hand'; $na{:}^{33}$ 'rice field' | $p^hɔ{:}^{41ʔ}$ 'father'; $raj^{41ʔ}$ 'dry field' | $na{:}m^{453ʔ}$ 'water'; $ma{:}j^{453ʔ}$ 'wood' | $mi{:}t^{41}$ 'knife'; $liat^{41}$ 'blood' | $nok^{45}$ 'bird'; $lak^{45}$ 'to steal' |

**Table 5.** Gedney's tone-box chart (Gedney [1972] 1989, p. 202) applied to the Chiang Rai variety of Northern Thai.

| Initials at Time of Tonal Splits | Proto-Tai Tones | | | | |
|---|---|---|---|---|---|
| | **A** | **B** | **C** | **DL** | **DS** |
| Voiceless friction sounds, *s-, *hm-, *ph-, etc. | $hu{:}^{24}$ 'ear'; $k^ha{:}^{24}$ 'leg' | $k^haj^{22}$ 'egg'; $p^ha{:}^{22}$ 'to split' | $k^ha{:}^{44ʔ}$ 'to kill'; $sia^{44ʔ}$ 'shirt' | $k^ha{:}t^{22}$ 'torn'; $ŋiak^{22}$ 'gums' | $mat^{24}$ 'flea'; $p^hak^{24}$ 'vegetable' |
| Voiceless unaspirated stops, *p-, *t-, *k-, etc. | $pi{:}^{24}$ 'year'; $ta{:}^{24}$ 'eye' | $pa{:}^{22}$ 'forest'; $kaj^{22}$ 'chicken' | $kaw^{44ʔ}$ 'nine'; $tom^{44ʔ}$ 'to boil' | $pɔ{:}t^{22}$ 'lung'; $tɔ{:}k^{22}$ 'to pound' | $kop^{24}$ 'frog'; $tap^{24}$ 'liver' |
| Glottal, *ʔ-, *ʔb-, etc. | $bin^{335}$ 'to fly'; $dɛ{:}ŋ^{335}$ 'red' | $da{:}^{22}$ 'to scold'; $ba{:}^{22}$ 'shoulder' | $ba{:}^{44ʔ}$ 'crazy'; $ba{:}n^{44ʔ}$ 'village' | $dɛ{:}t^{22}$ 'sunshine'; $ʔa{:}p^{22}$ 'to bathe' | $bet^{24}$ 'fishhook'; $ʔok^{24}$ 'chest' |
| Voiced, *b-, *m-, *l-, *z-, etc. | $mɨ{:}^{335}$ 'hand'; $na{:}^{335}$ 'rice field' | $pɔ{:}^{31}$ 'father'; $haj^{31}$ 'dry field' | $nam^{53ʔ}$ 'water'; $ma{:}j^{53ʔ}$ 'wood' | $mi{:}t^{31}$ 'knife'; $liat^{31}$ 'blood' | $nok^{35}$ 'bird'; $lak^{35}$ 'to steal' |

Synchronically, Standard Thai (Table 4) keeps the primary tonal splits between historical voiceless sounds and voiced sounds, which can be shown as in Sets 123-4, on proto-Tai tones *BCD, but has a secondary two-way split as Sets 1-234 on tone *A. Based on some historical documents of Thai, Tingsabadh (2001) and Pittayaporn (2018) pointed out that this tonal split pattern in tone *A in Standard Thai was the result of the original three-way split A1-23-4 to the later merger of tones A23 and A4, forming a secondary two-way split A1-234. This illustrates that the secondary tonal split was a kind of later change in relation to the primary tonal split. Liao (2016a, pp. 88–89; 2016b, pp. 168–72) illustrated this diachronic order in detail, as summarized at a glance in Liao (2022, p. 16): primary two-way split (e.g., *A > A1 and A2) > secondary three-way split (e.g., A1 > A1X and A1Y, and A1 ≠ A1Y ≠ A2) > secondary two-way split (e.g., A1Y merging into A2, and A1X ≠ A1Y = A2). In addition to this, a secondary merger took place in Standard Thai, in which tones B4 and

C123 merged, so that Standard Thai has five distinct tones (in the case of treating checked syllable tones to be allotones).

As for Northern Thai (Table 5), like Standard Thai, it only has a secondary tonal split in the original upper A tone, but the split is in a different position, appearing as A12-34. In addition, it does not have a horizontal tonal merger (which refers to mergers across A, B, C, and D tones), so it has six distinct tones. By comparing the tonal split patterns of Northern Thai and Standard Thai, we can understand why Gedney's formulation of the Southwestern Tai tones needs four sets of initial consonants on horizontal lines, because, in the case of A, there are splits of 1-234 (Standard Thai), 12-34 (Northern Thai), and 123-4 (both Standard and Northern Thai) for the other tones. In short, the examples of Standard Thai and Northern Thai show that Gedney's tone-box formulation ingeniously captures all tonal splits at a glance. More complex examples, such as Southern Thai with a three-way split in all five columns of *A/B/C/DL/DS (cf. Liao 2022, p. 6), can also be fully represented in this formulation.

However, as the variety of secondary tonal splits in Central Tai goes beyond Gedney's formulation, and the fixed testing lexical items in each box of Gedney's chart cannot deal with the so-called "voicing alternation" problem of the initial consonants between Southwestern/Central Tai and Northern Tai[5], such tone-box formulation has rarely been used in studies of the tonal systems of Tai dialects other than Southwestern Tai. In response to this problem, Liao (2022, cf. Liao 2016b, p. 211) proposed an integrated Tai tone-box scheme to capture all the tonal distinctions in any Tai variety and to solve the problems involved in various irregular tonal correspondences (including "voicing alternation"). The "main chart" of this scheme and its test etyma are shown in Tables 6 and 7, respectively.

**Table 6.** Integrated Tai (Kam-Tai) tone-box chart (Liao 2022, p. 17, cf. Liao 2016b, p. 211).

| Primary Initial Groups | Ultimate Laryngeal Natures at Time of Tonal Split | | | Tone Categories | | | | |
|---|---|---|---|---|---|---|---|---|
| | | | A | C | DS | B | DL |
| Proto-voiceless sounds (1) | (NT) Plain (1P) | (SWT/Saek/KS) Friction (1F) | Aspirated sounds (1A) | A1A | C1A | DS1A | B1A | DL1A |
| | | | Continuant sounds (1C) | A1C | C1C | DS1C | B1C | DL1C |
| | | Unaspirated stops (1U) | | A1U | C1U | DS1U | B1U | DL1U |
| | Glottal sounds (1G) | | | A1G | C1G | DS1G | B1G | DL1G |
| Proto-voiced sounds (2) | Plosives + continuants (2) | | | A2 | C2 | DS2 | B2 | DL2 |

There are two key points to this design. First, the maximum number of initial consonant sets that conditioned possible tonal splits in the Kam-Tai languages is displayed in this integrated Tai (Kam-Tai) tone-box chart as five, which the author called the "ultimate laryngeal natures at time of tonal split". They are aspirated sounds, such as *pʰ-, *tʰ-, *kʰ-, *kʰʷ-, *pʰʲ-, and *h-, voiceless (pre-aspirated) continuant sounds, such as *ʰm-, *ʰl-, *ʰŋ, and *ɬ-, unaspirated stops, such as *p-, *t-,*k-, *pl-, and *kw-, glottalized sounds, such as *ʔb-, *ʔd-, *ʔj-, *ʔw-, and *ʔ-, and voiced sounds, such as *b-, *d-, *n-, *ɟ-, and *ɣ-.[6] The first four of these five sets are all contained in proto-voiceless initials (the upper series), and the first two are contained in the voiceless friction sounds of Gedney's formulation. Some varieties of Central Tai and Yongnan Zhuang (belonging to Northern Tai) have a secondary tonal split between voiceless aspirated sounds and originally voiceless continuants on an original upper-register tone, e.g., kʰa:³¹ 'leg' and ma:⁵³ 'dog' in Debao Ma'ai Yang Zhuang both have initials belonging to Gedney's "friction sounds", but have tonal contrast (vs. Thai kʰa:²⁴ and ma:²⁴, having the same tone). Therefore, a distinction between voice-

less aspirated sounds and originally voiceless continuants needs to be made in this integrated chart[7].

**Table 7.** Test etyma for all Tai subgroups in the main chart of the integrated Tai tone-box scheme (Liao 2022, p. 18).

|  | **A** | **C** | **DS** | **B** | **DL** |
|---|---|---|---|---|---|
| 1A | *pra:$^A$ 'rocky hill/cliff'<br>*χiŋ$^A$ 'ginger'<br>*tri:l$^A$ 'stone' | *χɨ:n$^C$ 'upward'<br>*k.raj$^C$ 'fever'<br>*tram$^C$ 'to chop' | *prak$^D$ 'vegetable'<br>*krok$^D$ 'six'<br>*trep$^D$ 'hail' | *qraj$^B$ 'egg'<br>*pra:$^B$ 'to split'<br>*χɣaw$^B$ 'knee' | *pra:k$^D$ 'forehead'<br>*tra:p$^D$ 'to carry (a load)'<br>*xe:k$^D$ 'guest' |
| 1C | *$^h$na:$^A$ 'thick'<br>*$^h$ma:$^A$ 'dog'<br>*ɬa:w$^A$ 'girl' | *ɬom$^C$ 'sour'<br>*$^h$na:$^C$ 'face'<br>*$^h$ɲa:$^C$ 'grass' | *$^h$nak$^D$ 'heavy'<br>*$^h$mat$^D$ 'flea'<br>*ɬak$^D$ 'color/tattoo' | *ɬi:$^B$ 'four'<br>*$^h$nɣɤ:j$^B$ 'tired'<br>*$^h$mɣaɯ$^B$ 'new' | *$^h$ma:k$^D$ 'fruit'<br>*ɬa:p$^D$ 'cockroach'<br>*$^h$no:k$^D$ 'hump' |
| 1U | *tu:$^A$ 'door'<br>*kiŋ$^A$ 'to eat'<br>*pow$^A$ 'crab' | *kaw$^C$ 'nine'<br>*taŋ$^C$ 'to erect'<br>*pa:$^C$ 'aunt' | *tap$^D$ 'liver'<br>*kop$^D$ 'frog'<br>*pet$^D$ 'duck' | *kaj$^B$ 'chicken'<br>*tam$^B$ 'low'<br>*pɣaw$^B$ 'to blow' | *ko:t$^D$ 'to hug'<br>*pa:k$^D$ 'mouth'<br>*ka:t$^D$ 'leaf mustard' |
| 1G | *ʔdram$^A$ 'black'<br>*ʔbin$^A$ 'to fly'<br>*ʔjɣa:$^A$ 'medicine' | *ʔba:n$^C$ 'village'<br>*ʔdaj$^C$ 'to get'<br>*ʔo:j$^C$ 'sugarcane' | *ʔdrip$^D$ 'raw'<br>*ʔbup$^D$ 'concave'<br>*ʔik$^D$ 'chest' | *ʔbwa:$^B$ 'shoulder'<br>*ʔda:$^B$ 'to scold'<br>*ʔju:$^B$ 'to be at' | *ʔo:k$^D$ 'out'<br>*ʔdɨa:t$^D$ 'hot/boiled'<br>*ʔjɣa:k$^D$ 'hungry/to want' |
| 2 | *na:$^A$ 'rice field'<br>*ɣwa:j$^A$ 'buffalo'<br>*ga:$^A$ 'stuck' | *dɣu:ŋ$^C$ 'stomach'<br>*ma:$^C$ 'horse'<br>*li:n$^C$ 'tongue' | *ʼC.nok$^D$ 'bird'<br>*mot$^D$ 'ant'<br>*lak$^D$ 'to steal' | *bo:$^B$ 'father'<br>*gu:$^B$ 'pair'<br>*da:$^B$ 'river/wharf' | *liat$^D$ 'blood'<br>*ɟia:k$^D$ 'rope'<br>*C.ra:k$^D$ 'root' |

Second, in the integrated Tai tone-box chart, each initial set according to different laryngeal features that conditioned a possible tonal split in history is given a fixed name by using the combination of the number (1 for original voiceless and 2 for original voiced initials following the tradition of Tai tone studies) and abbreviations of those laryngeal features at the time of the tonal split, such as 1A for the voiceless aspirated initials and 1C for voiceless continuants (pre-aspirated sonorants + lateral *ɬ). By combining the tonal categories preceding these laryngeal sets, each tone-box is given a fixed name, such as A1A, A1C, A1U, A1G, A2, B1A, B1C, B1U, B1G, B2, etc. The order of the tonal categories in this chart is A, C, DS, B, and DL. This is because scholars in China are generally used to ordering Tai tones according to the order of A1 (1) A2 (2), C1 (3), C2 (4), B1 (5), B2 (6), DS1 (7), DS2 (8), DL1 (9), and DL2 (10) to correspond to the ordering of the LMC tones (as demonstrated in Table 2), and because, in most Tai varieties, the short vowel tonal category (*DS) in checked syllables tends to have the same split pattern as tone *C, while the long vowel tonal category (*DL) tends to have the same split pattern as tone *B (as seen in the tonal split patterns of several Tai varieties in Tables 8–10). Thus, C and DS, and B and DL are arranged together, making it easier to present such patterns. However, as the author indicated when he proposed this tone-box scheme, this ordering is not compulsory, and any investigator can follow the conventional order to reorder the horizontal columns of the integrated Kam-Tai tone-box as of A, B, C, DL, and DS, especially for the Kam-Sui branch, where this order makes more sense (as seen from the Southern Kam tonal split pattern displayed in Table 11) (cf. Liao 2022, pp. 22–23, 36). Tables 8–11 show that secondary tonal splits on original upper-register tones are a common Kam-Tai tone trait.

**Table 8.** Two-way split pattern in Debao Urban Yang Zhuang (Central Tai).

| | A | C | DS | B | DL |
|---|---|---|---|---|---|
| 1A | $p^hja:^{453}$ 'rocky hill' $k^h\partial\eta^{453}$ 'ginger' $t^h\partial n^{31}$ 'stone' | $k^h\upsilon n^{24?}$ 'upward' $k^hjaj^{24?}$ 'fever' $t^ham^{24?}$ 'to chop' | $p^hjak^{45}$ 'vegetable' $k^hjɔ:k^{45}$ 'six' $t^hap^{45}$ 'hail' | $k^hjai^{33}$ 'egg' $p^ha:^{33}$ 'to split' $k^haw^{33}$ 'knee' | $p^hja:k^{33}$ 'forehead' $t^ha:p^{33}$ 'to carry' $k^he:k^{33}$ 'guest' |
| 1C | $na:^{453}$ 'thick' $ma:^{453}$ 'dog' $ɬa:w^{453}$ 'girl' | $ɬam^{24?}$ 'sour' $na:^{24?}$ 'face' $ɲaj^{24?}$ 'grass' | $nak^{45}$ 'heavy' $mat^{45}$ 'flea' $ɬak^{45}$ 'color/tattoo' | $ɬaj^{55}$ 'four' $nu:j^{55}$ 'tired' $mɔ:j^{55}$ 'new' | $ma:k^{55}$ 'fruit' $ɬa:p^{55}$ 'cockroach' $no:k^{55}$ 'hump' |
| 1U | $tɘw^{453}$ 'door' $kɘn^{453}$ 'to eat' $pɘw^{453}$ 'crab' | $kaw^{24?}$ 'nine' $taŋ^{24?}$ 'to erect' $pa:^{24?}$ 'aunt' | $tap^{45}$ 'liver' $kap^{45}$ 'frog' $pat^{45}$ 'duck' | $kaj^{55}$ 'chicken' $tam^{55}$ 'low' $paw^{55}$ 'to blow' | $ko:t^{55}$ 'to hug' $pa:k^{55}$ 'mouth' $ka:t^{55}$ 'leaf mustard' |
| 1G | $ʔdam^{31}$ 'black' $ʔban^{31}$ 'to fly' $ʔja:^{31}$ 'medicine' | $ʔba:n^{24?}$ 'village' $ʔdaj^{24?}$ 'to get' $ʔo:j^{24?}$ 'sugarcane' | $ʔdɘp^{45}$ 'raw' $ʔbʊp^{45}$ 'concave' $ʔak^{45}$ 'chest' | $ʔba:^{33}$ 'shoulder' $ʔda:^{33}$ 'to scold' $ʔjɘw^{33}$ 'to be at' | $ʔo:k^{33}$ 'out' $ʔdu:t^{33}$ 'hot/boiled' $ʔja:k^{33}$ 'hungry' |
| 2 | $na:^{31}$ 'rice field' $va:j^{31}$ 'buffalo' $ka:^{31}$ 'stuck' | $to:ŋ^{213?}$ 'stomach' $ma:^{213?}$ 'horse' $lɘn^{213?}$ 'tongue' | $nɔ:k^{33}$ 'bird' $mɔ:t^{33}$ 'ant' $lak^{33}$ 'to steal' | $po:^{33}$ 'father' $kɘw^{33}$ 'pair' $ta:^{33}$ 'river' | $lu:t^{33}$ 'blood' $tɕɤ:k^{33}$ 'rope' $la:k^{33}$ 'root' |

**Table 9.** Three-way split pattern in Baoxu Zhuang (Central Tai) (cf. Liao 2016b, p. 170).

| Primary Initial Groups | Ultimate Phonetic Natures at Time of Tonal Splits | | | Tone Categories | | | | |
|---|---|---|---|---|---|---|---|---|
| | | | | A | C | DS | B | DL |
| Proto-voiceless sounds (1) | Plain (1P) | Friction (1F) | Aspirated sounds (1A) | 45 | 13? | 13 | 34 | 34 |
| | | | Continuant sounds (1C) | 45 | 13? | 13 | 34 | 34 |
| | | Unaspirated stops (1U) | | 53 | 34? | 34 | 33 | 33 |
| | Glottal sounds (1G) | | | 53 | 34? | 34 | 33 | 33 |
| Proto-voiced sounds (2) | Plosives + Continuants (2) | | | 31 | 11? | 11 | 33 | 33 |

**Table 10.** Three-way split pattern in Xialeng Zhuang (Yongnan Zhuang) (Liao 2022, p. 27).

| Primary Initial Groups | Ultimate Phonetic Natures at Time of Tonal Splits | | | Tone Categories | | | | |
|---|---|---|---|---|---|---|---|---|
| | | | | A | C | DS | B | DL |
| Proto-voiceless sounds (1) | Plain (1P) | Friction (1F) | Aspirated sounds (1A) | 33 | 13 | 55 | 53 | 53 |
| | | | Continuant sounds (1C) | 55 | 35 | 55 | 13 | 13 |
| | | Unaspirated stops (1U) | | 55 | 35 | 55 | 13 | 13 |
| | Glottal sounds (1G) | | | 33 | 13 | 55 | 53 | 53 |
| Proto-voiced sounds (2) | Plosives + Continuants (2) | | | 31 | 11 | 33 | 53 | 53 |

**Table 11.** Three-way split pattern in the Zhanglu Variety of Southern Kam (Kam-Sui) (Liao 2022, p. 36).

|  |  | **A** | **B** | **C** | **DL** | **DS** |
|---|---|---|---|---|---|---|
| 1F | (1A) | 55 | 53 | 323 | 323/24 | 55 |
|  | (1C) |  |  |  |  |  |
| 1U/G | (1U) | 35 | 453 | 13 | 13 | 35 |
|  | (1G) |  |  |  |  |  |
|  | 2 | 212 | 33 | 31 | 31 | 21 |

The tonal split patterns in the four Kam-Tai dialects shown in Tables 8–10 are all different from each other in terms of the secondary split. The number of initial sets, the number of ways of tone differentiation (two-way or three-way), and the number of distinct tones required for each dialect show a complex relationship that is not necessarily proportional. Debao Urban Yang Zhuang only has a two-way split and six tones, but it requires four sets of initials, as its tonal split pattern is 1A-1C/U-1G-2. The other three languages have seemingly more complicated tonal split patterns than Debao Urban Yang Zhuang in terms of three-way split and the tone numbers being seven (Baoxu and Xialeng Zhuang) and nine (Southern Kam), but the required numbers of sets of initials are only three, 1AC-1UG-2 (for both Baoxu Zhuang and Southern Kam), and four, 1A-1CU-1G-2 (Xialeng Zhuang), respectively.

For two Central Tai varieties, the patterns 1A-1C/U-1G-2 (Debao Yang Zhuang in Table 8) and 1AC-1UG-2 (Baoxu Zhuang in Table 9) are combined to require all the five sets 1A-1C-1U-1G-2 of the integrated Tai tone-box chart. In contrast, the other Kam-Tai subgroups require fewer initial sets if only focusing on their internal tone systems, namely three sets for Northern Zhuang (1ACU-1G-2) (cf. Liao 2016b, p. 209), Saek (Liao 2022, p. 34), and Kam-Sui (Liao 2022, p. 36), four sets for Southwestern Thai (1AC-1U-1G-2, as in the Standard Thai and Northern Thai cases in Tables 4 and 5) and Yongnan Zhuang (1A-1C/U-1G-2, as in Xialeng Zhuang in Table 10). Therefore, in terms of integrity, five sets of initial consonants are required to be compatible with the complex tonal split patterns of the entire Kam-Tai languages. In other words, Central Tai is the subgroup with the most complex tonal split patterns among all Kam-Tai languages. From the studied Kra-Dai languages, in terms of tonal split patterns, the two branches Tai and Kam-Sui are already the most complex. The other branches or subbranches, namely Kra, Be, and Hlai, have a much simpler pattern, which is generally similar to common Sinitic languages' upper-lower register split pattern. Thus, the integrated Kam-Tai tone-box, which contains the initial sets of these five laryngeal features, already contains the most complex tonal splitting conditions for all Kam-Tai languages known.

As demonstrated, a tone-box chart is not only a handy tool for studying the complexities of tone systems, but also a framework for presenting tonal split patterns at a glance. In previous studies, we saw that the tone systems of some specific Lingnan Sinitic dialects (i.e., various varieties of Yue Chinese and Pinghua that have been in close contact with Kam-Tai for a millennium) have similar secondary tonal split patterns to those of Kam-Tai languages. In order to present the tonal split patterns of these Lingnan Sinitic languages visually, modeled on the Kam-Tai tone-box framework, I assembled an integrated Sinitic tone-box chart, as shown in Table 12.

**Table 12.** An integrated (Lingnan) Sinitic tone-box chart.

| EMC Initial Voicing | Ultimate Laryngeal Natures at Time of Tonal Split | Sinitic Tonal Categories | | | | |
|---|---|---|---|---|---|---|
| | | A 平 *píng* | B 上 *shǎng* | C 去 *qù* | DL 長入 long *rù* | DS 短入 short *rù* |
| 清 (clear) Voiceless (1) | 次清 (semi-clear) Aspirated surd (1A) | A1A 方 清 溪 | B1A 粉 體 起 | C1A 派 透 去 | DL1A 法 塔 客 | DS1A 匹 七 曲 |
| | 全清 (clear) Unaspirated surd (1U) | A1U 幫 知 乖 | B1U 審 古 影 | C1U 變 對 見 | DL1U 百 節 覺 | DS1U 北 德 屋 |
| 濁 (muddy) Voiced (2) | 次濁 (semi-muddy) Sonorants/Liquid (2S) | A2S 明 來 疑 | B2S 武 朗 五 | C2S 望 弄 用 | DL2S 滅 納 月 | DS2S 物 入 日 |
| | 全濁(muddy) Obstruent (stops/fricatives) (2O) | A2O 旁 唐 群 | B2O 坐 斷 近 | C2O 病 定 具 | DL2O 白 絕 達 | DS2O 讀 十 及 |

This Sinitic tone-box chart is firstly based on the determination of the upper limit of the set of initial laryngeal features that determine possible tonal split in these Sinitic languages, inspired by Kam-Tai tone-box theories. It is also a framework designed in a way that is compatible with the tradition of Sinitic tonology for facilitating the discussion of tonal split patterns as an areal trait in Lingnan languages in the next section. The four horizontal lines in this chart are arranged according to the four sets of initial consonants with different laryngeal features in traditional Chinese phonology, namely voiceless aspirated obstruents, voiceless unaspirated obstruents, voiced sonorants, and voiced obstruents. The three Chinese characters in each box were selected from the list of characters in the GuangYun (廣韻), the Chinese rime dictionary that was compiled from 1007 to 1008 by the Northern Song court, and were placed in the box according to their initial consonant category (horizontally) and tonal category (vertically). In the checked syllables (the DL and DS columns), the Chinese characters were also assigned respectively according to their rime's long vowel (corresponding to one of the vowels a, ɛ, œ, ɔ, i, u, y in Guangzhou/Hong Kong Cantonese) or short vowel (corresponding to one of the vowels ɐ, ɪ, ʊ, ɵ in Guangzhou/Hong Kong Cantonese), because the determination of their "long" and "short" status was quite consistent in their conditioning role in the possible horizontal tonal split in checked syllables in the Yue and Pinghua dialects, which was suggested to be a trait diffused from Kam-Tai, as mentioned. Although this chart is theoretically able to capture most tonal distinctions of all Sinitic languages as long as they follow regular tone rules, I am cautiously limiting its application to the Sinitic languages in the core Lingnan area and its surrounding areas for the time being, as it might not be necessary for Sinitic languages elsewhere to require such a formulation.

## 3. Similarities of Tonal Behavior in Lingnan Languages

In the subsections of this section, we examine the tonal split patterns and certain similarities in tonal behavior between the Siniti and Kam-Tai languages of the core Lingnan area from three perspectives: upper-register secondary tonal splits, lower-register secondary tonal splits, and upper-register tone associated with sonorant initials, in order to trace their historical contact and mutual influence.

### 3.1. Upper Register Secondary Tonal Split as a Kam-Tai Origin Trait

We devoted considerable space in Section 2 to the tonal development of the Kam-Tai languages in Lingnan and MSEA in order to distill and summarize the tonal split patterns that are of the areal traits of this region. By virtue of the convenience of the tone-box frameworks, it is clear that the feature of potential secondary splits in the upper register of the primary tonal split is a trait of the Kam-Tai languages. Furthermore, this trait, together with the previously mentioned potential split according to vowel length in checked sylla-

bles, has diffused into some of the Lingnan Sinitic languages, particularly some Yue and Pinghua dialects. Let us now take a look at some of the details of the distribution of these traits in the two major language groups Kam-Tai and Sinitic.

First, there are the Kam-Tai languages. While the various secondary tonal split patterns of Kam-Tai languages are illustrated in Section 2, their status as a "trait", even within Kam-Tai languages themselves, needs to be further clarified. Crucially, the criterion for the number of languages, or rather language families, in a linguistic area has been the subject of controversial scholarly discussion, such as three or more languages (Thomason 2001, p. 99), three languages not belonging to the same family (Schaller 1975, p. 58), and more than two languages (Campbell 1985, p. 25). However, as Campbell (2006, pp. 7–8; 2017, p. 25) points out that even the Balkans, probably the most well-established sprachbund, involves only Indo-European languages, and that the requirements for the number of languages or language families in a sprachbund are not really a focus of attention, we can conclude that at least two or more languages without a genetically close relationship is basically a requirement. Moreover, Szeto and Yurayong (2022, p. 44) pointed out that western Lingnan already has languages from five language families, so the requirement for Lingnan to establish a sprachbund is even less of a problem. On the face of it, one could certainly argue that some of the tonal behaviors as features shared by languages from the Kam-Tai and Sinitic languages in Lingnan are the result of diffusion. However, when we only focus on the Kam-Tai languages, which belong to the same Kra-Dai family, the sharing of a trait, such as the secondary tonal split of the original high-register tone commonly found in Kam-Tai languages, first appeared to be an inherent feature of Kam-Tai, rather than a trait acquired through diffusion. However, as we have demonstrated that at least tonogenesis and primary tonal split are indeed traits diffusing from Sinitic languages in history, the determination of abundant secondary tonal split patterns as "an inherent feature" becomes problematic. A further examination of the specific distribution of the different patterns of secondary tonal splits in various Kam-Tai languages themselves shows that they are indeed a trait, as demonstrated below.

According to Liao and Tai (2019), we have already arrived at that, among the five terminal subgroups of Tai, as shown in Figure 2, Northern Zhuang, Yongnan Zhuang, and Saek are grouped together to form Northern Tai, while Central Tai and Southwestern Tai are grouped together to form Southern Tai. Northern Tai and Southern Tai form the Tai subbranch of Kam-Tai. Meanwhile, the Kam-Sui languages are another subbranch distinct from Tai. Further, Kam-Sui and Tai have been commonly placed under Kam-Tai, which is mostly regarded as a first-order primary branch under Kra-Dai. Not proportional to the above affinities is the fact that, as Northern Zhuang did not develop aspirated initials, the language varieties belonging to it lack possible secondary tonal split patterns conditioned by aspirated sounds (1A), which is, however, shared by some more distantly related Yongnan Zhuang and Central Tai language varieties, as shown in the tonal split patterns of Debao Urban Yang Zhuang (Central Tai) and Xialeng Zhuang (Yongnan Zhuang) demonstrated in Tables 8 and 10 above. In addition, in Northern Zhuang varieties, unaspirated voiceless stops (1U) and original voiceless frictions (i.e., 1F) conditioned tonal split together everywhere, so these two sets were grouped together to be the same initial set 1P in Northern Zhuang (referring to the integrated Tai tone-box in Table 6). However, that 1F and 1U being treated as two individual sets is shared by more distantly related language varieties, such as Southern Kam and Mak of Kam-Sui, Saek, Baoxu Zhuang, and Leiping Ping Zhuang of Central Tai, and Standard Thai, Northern Thai, Laos, and Southern Thai of Southwestern Tai. Note that Saek's tonal split pattern is very close to those of Standard Thai and Southern Kam (cf. Liao 2022, pp. 34–36). It is also important to keep in mind that, in the languages in all the terminal subgroups of the Tai and Kam-Sui branches, there may be languages that retain the most primitive primary tonal split pattern in all the proto-tones—for example, non-Southern Kam and non-Mak Kam-Sui languages, about half of the Central Tai and Yongnan Zhuang dialects, most of the Northern Zhuang dialects, and some of the Southwestern dialects (such as Tai Lue and Shan)—and preserve the primary tonal

split patterns; i.e., they have no secondary tonal split patterns. Moreover, it is possible that, in a single Tai language (determined based on the principles of intelligibility, similarity of wordlists, and speaker attitudes), such as Yang Zhuang, there are many dialects without any secondary tonal split patterns (e.g., Jingxi and Napo Yang Zhuang), but there is also a cluster of dialects with complex secondary tonal split patterns (e.g., various Debao Yang Zhuang varieties and the Hurun variety of Jingxi Yang Zhuang) (cf. Liao 2016b, p. 257 ff.). Therefore, such secondary tonal split patterns have to be considered as an areal trait across many of the more distantly related Kam-Tai language varieties. Note that this trait has also penetrated MSEA with the migration of Southwestern Tai and is not restricted to Lingnan, but it is not found in any non-Tai languages on the Indo-China Peninsula, the heartland of MSEA. The details of this trait in the Kam-Tai languages demonstrated or mentioned above, together with some other Tai varieties in Liao's (2016b, p. 257 ff.) study of Tai tonal development, are condensed in Figure 4 so that the readers can understand the nature of it as a diffused "trait" at a glance.

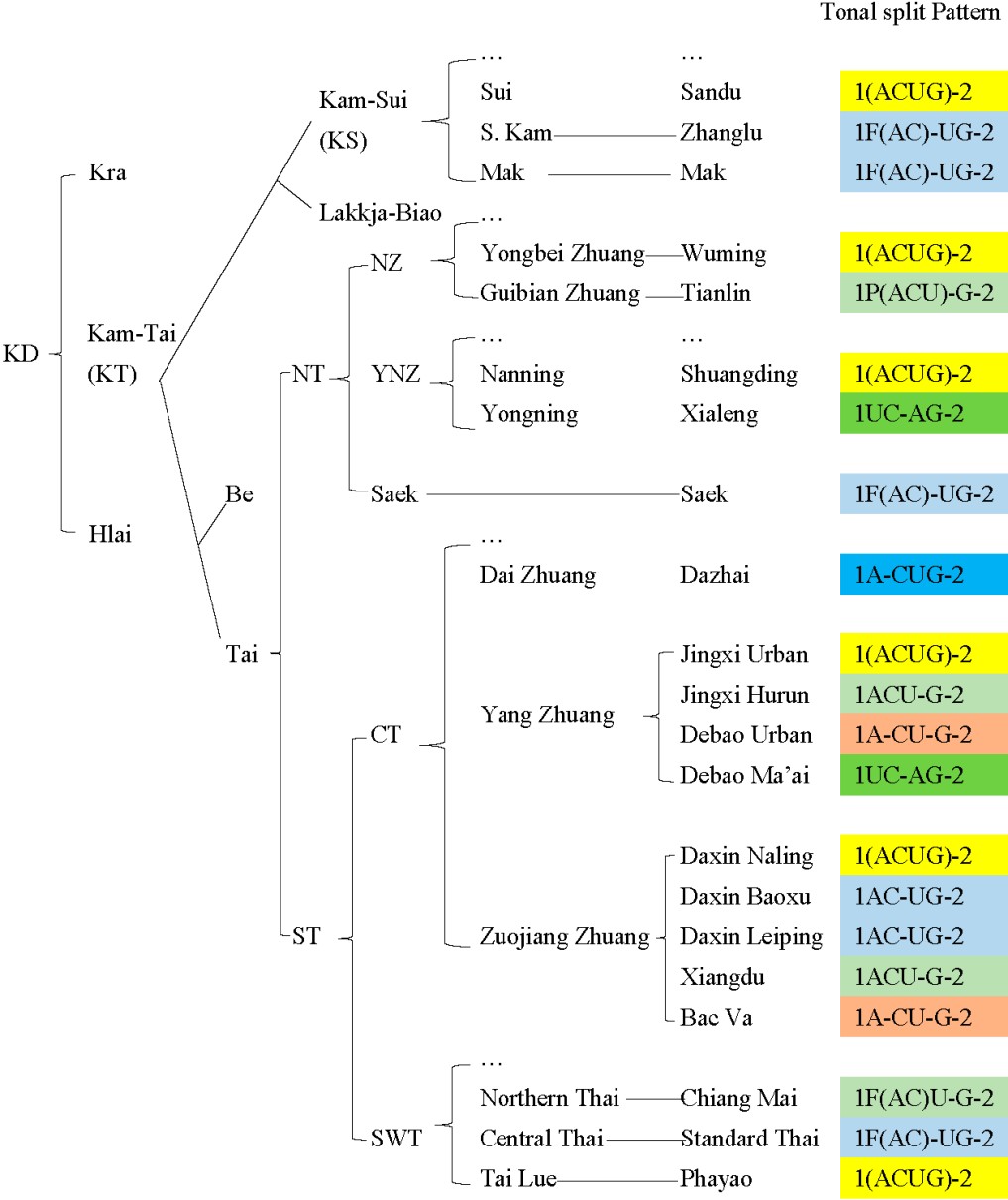

**Figure 4.** Tonal split patterns as a trait distributed in the Kam-Tai languages.

In Figure 4, the language groups and language varieties are arranged by their respective affiliations within the Kam-Tai family according to their close or distant genetic relationship, and each language variety is labeled with its maximum tonal split patterns, i.e., how many sets of initial consonants are needed to capture all the tonal split patterns of that dialect, with each set of initial consonants being separated by a dash to indicate a horizontal line (in a particular tone-box chart). The same pattern is highlighted in the same color, e.g., yellow is the primary tonal split pattern, and those language varieties that maintain the primary tonal split pattern are uniformly represented by the pattern of 1(ACUG)-2, but for others where a secondary split pattern occurs, the set name of the initials of the tone-box chart of that terminal Tai subgroup is followed by parentheses in which the set names of the integrated tone-box chart are filled. For example, Tianlin Zhuang in Northern Zhuang, whose maximum tonal split pattern is 1P(ACU)-G-2, where 1P refers to "plain voiceless sounds" (excluding glottalized sounds in the upper register initials) in the Northern Tai tone-box chart (Liao 2016b, p. 209), and (ACU) in the parentheses is the set names of the first three horizontal lines of the integrated Kam-Tai tone-box (Table 6) corresponding to Set 1P for Northern Zhuang varieties.

However, we should be aware that pattern 1P(ACU)-G-2 of Tianlin Zhuang is not substantially different from 1ACU-G-2 of Jingxi Hurun Yang Zhuang and Xiangdu Zuojiang Zhuang, as well as 1F(AC)U-G-2 of Northern Thai, as they are all highlighted in the same salmon pink color. They differ in labeling only because the distinction between sets within P and F is unnecessary for Northern Zhuang (to which Tianlin Zhuang belongs) and Southwestern Tai (to which Northern Tai belongs), respectively, and the distinction of A, C, and U is necessary for Central Tai (to which Hurun Yang Zhuang belongs). From this figure, we can conclude that the tonal split trait is manifested in the Kam-Tai languages in the following ways.

(A) The primary tonal split pattern is found in every terminal subgroup of Tai, except for Saek, which is a single language only including a small number of varieties. This pattern actually refers to a retention of the post-proto-Tai feature;

(B) Northern Zhuang's maximum pattern 1P(ACU)-G-2, secondary tonal split conditioned by 1G alone, is also found in some Central Tai and Southwestern Tai languages;

(C) There is a special pattern, 1A-CUG-2, in Dai Zhuang of Central Tai, with the secondary tonal split conditioned by aspirated initials (1A) alone. According to Liao (2016b, pp. 134, 320), the Huashan variety of Min Zhuang (Central Tai) also has this pattern;

(D) Secondary tonal split conditioned by unaspirated stops (1U) and glottalized sounds (1G) together, 1F(AC)-UG-2 or 1AC-UG-2, is found in Kam-Sui, Central Tai, Saek, and Southwestern Tai, and this is the most widely spread pattern in terms of the secondary tonal split in Kam-Tai languages, both geographically and dialectally;

(E) Secondary tonal split conditioned by aspirated sounds (1A) and glottalized sounds (1G) together, 1UC-AG-2, is only found in Central Tai and Yongnan Zhuang, which are geographically close to each other in southcentral–southwestern Guangxi;

(F) Secondary tonal split conditioned by aspirated sounds (1A) or glottalized sounds (1G), but not always together on all tones that have secondary tonal split, form a pattern of 1A-CU-G-2, is only found in Central Tai, but it is dispersed in language varieties in different Central Tai languages, namely Yang Zhuang and Zuojiang Zhuang (as well as Min Zhuang according to Liao 2016b, pp. 134, 320), which contain more patterns in their respective dialectal varieties;

(G) Taken together, as mentioned in Section 2, Central Tai is the most internally diverse terminal subgroup of all Kam-Tai languages in terms of tonal split patterns. This is understandable in terms of its geographical distribution, as it is situated south of Kam-Sui and Northern Tai, west and south-west of Yongnan Zhuang, and north-east of Southwestern Tai in a central transition zone. From these details, we are confident in considering the upper-register secondary tonal split patterns as a complex

phonological trait in the Kam-Tai languages, even not including its diffusion into the Sinitic languages.

Next, let us move on to consider how the trait of a secondary tonal split diffused from Kam-Tai into Lingnan Sinitic languages, as well as from Sinitic to Kam-Tai, if applicable. We begin below by examining the two Sinitic dialects, Standard Cantonese (Guangzhou-Hong Kong) and the Luxu (蘆墟) variety of Binyang (賓陽) Pinghua in the integrated Sinitic tone-box chart in Tables 13 and 14, respectively, to see how the tonal split patterns from Kam-Tai have diffused to Lingnan Sinitic languages in a specific way[8].

**Table 13.** Tonal split patterns in Guangzhou–Hong Kong Cantonese.

| | A | B | C | DL | DS |
|---|---|---|---|---|---|
| 1A | 方*fɔŋ*⁵⁵ 'square'<br>清 *tʃʰɪŋ*⁵⁵'clear'<br>溪*kʰei*⁵⁵ 'stream' | 粉*fɐn*²⁵ 'powder'<br>體*tʰei*²⁵'body'<br>起*hei*²⁵ 'rise' | 派*pʰai*³³ 'send'<br>透*tʰɐu*³³ 'pierce'<br>去*hɵy*³³ 'go' | 法*fat*³³ 'law'<br>塔*tʰap*³³ 'tower'<br>客*hak*³³ 'guest' | 匹*pʰɐt*⁵⁵ 'equal'<br>七*tʃʰɐt*⁵⁵ 'seven'<br>曲*kʰʊk*⁵⁵ 'music' |
| 1U | 幫*pɔŋ*⁵⁵ 'help'<br>知*tʃi*⁵⁵ 'know'<br>乖*kʷai*⁵⁵ 'tractable' | 審 *ʃɐm*²⁵ 'interrogate'<br>古*ku*²⁵ 'to get'<br>影*jɪŋ*²⁵ 'shadow' | 變*pin*³³ 'change'<br>對*tɵy*³³'correct'<br>見*kin*³³ 'see' | 百*pak*³³ 'hundred'<br>節*tʃit*³³ 'festival'<br>覺*kɔk*³³ 'feel' | 北*pɐk*⁵⁵ 'north'<br>德*tɐk*⁵⁵ 'morality'<br>屋*ʔʊk*⁵⁵ 'room' |
| 2S | 明*mɪŋ*²¹ 'light'<br>來*lɔi*²¹ 'come'<br>疑*ji*²¹ 'suspect' | 武*mou*²³ 'military'<br>朗*lɔŋ*²³ 'bright'<br>五*ŋ̩*²³ 'five' | 望*mɔŋ*²² 'look'<br>弄*nʊŋ*²² 'handle'<br>用*jʊŋ*²² 'use' | 滅*mit*²² 'extinguish'<br>納*nap*²² 'accept'<br>月*jyt*²² 'month' | 物*mɐt*²² 'object'<br>入*jɐp*²² 'enter'<br>日*jɐt*²² 'date' |
| 2O | 旁*pʰɔŋ*²¹ 'side'<br>唐*tʰɔŋ*²¹ 'bright'<br>群*kʰʷɐn*²¹ 'group'<br><br>坐*tʃɔ*²² 'sit'<br>斷*tyn*²² 'broken'<br>近*kɐn*²² 'near' | 坐*tʃʰɔ*²³ 'sit'<br>斷*tʰyn*²³ 'broken'<br>近*kʰɐn*²³ 'near' | 病*pɛŋ*²² 'sick'<br>定*tɪŋ*²² 'certain'<br>具*kɵy*²² 'tool' | 白*pak*²² 'white'<br>絕*tʃyt*²² 'extremely'<br>達*tat*²² 'reach' | 讀*tʊk*²² 'read'<br>十*ʃɐp*²² 'ten'<br>及*kʰɐp*²² 'rich' |

**Table 14.** Tonal split patterns in Binyang Luxu Pinghua.

| | A | B | C | DL | DS |
|---|---|---|---|---|---|
| 1A | 方*fuŋ*³⁴ 'square'<br>清 *tsʰɐŋ*³⁴ 'clear'<br>溪*hei*³⁴ 'stream' | 粉*fən*³³ 'powder'<br>體*tʰei*³³ 'body'<br>起*hei*³³ 'rise' | 派*pʰai*⁵⁵ 'style'<br>透*tʰɔu*⁵⁵ 'pierce'<br>去*hu*⁵⁵ 'go' | 法*fap*³³ 'law'<br>塔*tʰap*³³ 'tower'<br>客*hak*³³ 'guest' | 匹*pʰɐt*⁵⁵ 'equal'<br>七*tsʰɐt*⁵⁵ 'seven'<br>曲*kʰɔk*⁵⁵ 'music' |
| 1U | 幫*paŋ*³⁴ 'help'<br>知*tsi*³⁴ 'know'<br>乖*kʷai*³⁴ 'tractable' | 審*səm*³³ 'interrogate'<br>古*kou*³³ 'to get'<br>影*ʔɐŋ*³³ 'shadow' | 變*pin*⁵⁵ 'change'<br>對*tu*⁵⁵ 'correct'<br>見*kin*⁵⁵ 'see' | 百*pak*³³ 'hundred'<br>節*tsit*³³ 'festival'<br>覺*tsak*³³ 'feel' | 北*pɐk*⁵⁵ 'north'<br>德*tɐk*⁵⁵ 'morality'<br>屋*ʔʊk*⁵⁵ 'room' |
| 2S | 明*mɐn*²¹³ 'light'<br>來*lai*²¹³ 'come'<br>疑*ɲi*²¹³ 'suspect' | 武*fou*²² 'military'<br>朗*løŋ*²² 'bright'<br>五*ŋɔu*²² 'five' | 望*muŋ*⁴² 'look'<br>弄*non*⁴² 'handle'<br>用*jon*⁴² 'use' | 滅*mit*⁴² 'extinguish'<br>納*nap*⁴² 'accept'<br>月*ɲut*⁴² 'month' | 物*fɐt*¹¹ 'object'<br>入*ɲap*¹¹ 'enter'<br>日*ɲɐt*¹¹ 'date' |
| 2O | 旁*puŋ*²¹³ 'side'<br>唐*tøŋ*²¹³ 'bright'<br>群*kʷən*²¹³ 'group' | 坐*tsou*²² 'sit'<br>斷*tun*²² 'broken'<br>近*kən*²² 'near' | 病*pɐŋ*⁴² 'sick'<br>定*tɐŋ*⁴² 'certain'<br>具*ku*⁴² 'tool' | 白*pak*⁴² 'white'<br>絕*tʃit*⁴² 'extremely'<br>達*tat*⁴² 'reach' | 讀*tɔk*¹¹ 'read'<br>十*səp*¹¹ 'ten'<br>及*tsap*¹¹ 'rich' |

As can be seen from the above two tables, the horizontal tonal split patterns of both Cantonese in Guangzhou/Hong Kong and Pingyang were neatly conditioned by the voic-

ing of initial consonants in history, i.e., they are equivalent to the primary tonal split of the Kam-Tai languages, so there are no secondary tonal split patterns in the upper register, as found in many Kam-Tai languages. However, both dialects have vertical tonal split patterns in checked syllables (columns DL and DS); that is, secondary tonal splits conditioned by vowel length in checked syllables. In particular, in Cantonese, this secondary split is only found in the upper register, which means that the items in the DL1(AU) and DS1(AU) boxes are systematically different in their tonal values. In contrast, Binyang Luxu Pinghua has splits in both the upper and lower registers, i.e., DL1, DL2, DS1, and DS2 all contain lexical items with different tones to the items in the other boxes. In previous studies, such patterns, especially the Cantonese one, have been regarded as an areal trait diffused from the neighboring Kam-Tai languages, as mentioned. In Table 13, the three characters in the Cantonese Tone-Box B2O have two tones each, marked with different colors according to their respective tonemes, and with the arrow which indicates the tonal merging direction, i.e., a special secondary tonal split occurs in the low-register B2O, which we will discuss in detail in Section 3.2, because in this section, we are only concentrating on the secondary tonal split of the upper register.

Next, we look at the tonal split patterns of another Pinghua dialect, Nanning Weizilu (位子渌) Pinghua, and another Yue dialect, Rongxian (容縣) Yue of Guangxi, as shown in Tables 15 and 16, summarized from de Sousa (Forthcoming) and Ran et al. (2015, p. 32) respectively.

**Table 15.** Tonal split patterns in the Weizilu variety of Pinghua in suburban Nanning.

| | A | B | C | DL | DS |
|---|---|---|---|---|---|
| 1A | 53 | 33 | 35 | 33 | 33 |
| 1U | 53 | 33 | 55 | 33 | 33 |
| 2S | 21 | 13 | 22 | 23 | 23 |
| 2O | 21 | 13 / 22 | 22 | 22 | 22 |

**Table 16.** Tonal split patterns in the Rongxian variety of Yue Chinese in eastern Guangxi.

| | A | B | C | DL | DS |
|---|---|---|---|---|---|
| 1A | 35 | 33 | 452 | 35 | 35 |
| 1U | 55 | 44 | 52 | 43 | 55 |
| 2S | 343 | 23 | 31 | 32 | 32 |
| 2O | 343 | 23 / 31 | 31 | 32 | 32 |

As shown in Tables 15 and 16, there are secondary tonal splits in the lower registers of both dialects, and this topic is also discussed in Section 3.2 because, here, we focus only on the splits of the upper register. In the patterns of Weizilu Pinghua shown in Table 15, there are no vertical secondary tonal splits according to vowel length on checked syllables, but a secondary tonal split due to the association of voiceless obstruents (including h-) is found in tone C1 (the upper *qù*); that is, the split between Box C1U and Box C1A. Such an upper-register secondary tonal split pattern is found in Weizilu Pinghua only in the upper tone C, but in Rongxian Yue (Table 16), we see a very similar pattern to that of Southern Kam, where each upper-register tone has a secondary tonal split, making this Sinitic dialect a rare one with nine distinct tones (while treating the tones of the checked syllables as allotones). Such upper-register secondary tonal split patterns are quite rarely found in Sinitic languages, but are very commonly found in Kam-Tai languages, as discussed previously.

Our judgment that secondary tonal split patterns in checked syllables and the upper register occurred first in Kam-Tai and diffused into Sinitic, rather than the other direc-

tion, is based on two pieces of evidence presented in the above discussion. One is that secondary tonal split patterns can be found in all Kam-Tai branches and are widespread, but in Sinitic languages, at present, they are only found in Pinghua and Yue, which are the closest to Kam-Tai in terms of language typology (cf. Matthews 2006; de Sousa 2015a, 2015b; Szeto 2019; Szeto and Yurayong 2021, 2022). The second is that, in Sinitic languages, only Pinghua and Cantonese have vowel length contrast, like in Kam-Tai languages, and there are only two contrasting laryngeal features, the semi-clear (次清)/aspirated surd (1A) and the clear (全清)/unaspirated surd (1U), in the original voiceless initials of the Sinitic languages. However, in Kam-Tai, there are four sets (1A, 1C, 1U, and 1G). This is why the Kam-Tai languages have more accessible laryngeal conditions for the occurrence of secondary tonal splits in the original upper-register tones. Simply by virtue of this, we conclude that the Lingnan Sinitic varieties, such as Weizilu Pinghua and Rongxian Yue, which have co-existed with Kam-Tai languages in the Lingnan region for around a thousand years, have received a trait from Kam-Tai languages in the pattern of the upper-register tonal splits.

As a result of the comparison of Kam-Tai and Sinitic tones, in this subsection, by using the frameworks of tone-box charts, we can summarize the following Lingnan tonal behavior traits involved in the original upper-register tones as in (1)–(2) below.

(1)    Areal Trait 1 of tonal behavior in Lingnan languages: The secondary tonal split is conditioned by vowel length on checked syllables spread from Kam-Tai languages to some Sinitic languages, such as some varieties of Cantonese and Southern Pinghua.

(2)    Areal Trait 2 of tonal behavior in Lingnan languages: The pattern of the secondary tonal split of the upper-register tones spread among various subgroups of Kam-Tai languages and diffused into some nearby Sinitic languages, such as some varieties of Nanning Weizilu Pinghua and Rongxian Yue.

Having established the properties of the above two traits, we turn to the areal typological issues associated with them. At this point, we first need to determine the diachronic order of the primary and secondary tonal splits and the respective triggers that gave rise to them. We will show, in Section 3.3, that the merger of original voiceless and voiced continuants was indeed the trigger of the primary tonal split in Kam-Tai, in contrast with Sinitic, in which the devoicing of original voiced obstruents was the trigger of the register tonal split. Thus, for the triggers of the secondary tonal split, the laryngeal features of original voiceless continuants and original voiceless obstruents are excluded from Kam-Tai and Sinitic, respectively, while the remaining laryngeal features are considered to be the triggers of secondary tonal split. In the inherited etyma of the Sinitic languages (excluding Kam-Tai stratum/loans), because there are only two sets of laryngeal features (aspirated and unaspirated surds) in the upper-register initials, there is only one possibility for the secondary tonal split in the upper register, namely the condition of the set of aspirated obstruents (1A). In contrast, the Kam-Tai languages have three possible conditions for secondary splits, namely aspirated sounds (1A), unaspirated stops (1U), and glottalized sounds (1G). This is why secondary tonal split patterns of the Kam-Tai languages are likely the most complex in this area[9]. Finally, we are able to determine three areal types of tones in Lingnan Sinitic and Kam-Tai, as summarized in (3)–(5).

(3)    Lingnan languages' phonological (tonological) Areal Type 1: The upper limit for the laryngeal features in the upper register of initial consonants that may have conditioned the further possible secondary tonal splits is three, namely aspirated (1A), unaspirated (1U), and glottalized (1G), contrasting with original voiceless continuant (1C), which is suggested to be the cause of the primary tonal split (see Section 3.3).

(4)    Lingnan languages' phonological (tonological) Areal Type 2: In a single language variety, due to limitations in terms of tone load-bearing capacity, the upper limit for further splits in the high-register tone is two, a primary high-register tone and a secondary high-register tone (whether or not this splitting tone eventually merges into the counterpart low-register tone to form another two-way split pattern).

This means that, although the maximum number of initial sets that conditioned the high register secondary tonal split is three, only one (1A or 1G) of these sets alone, or two (1 A and 1G, or 1U and 1G) of them together, conditioned the same secondary tonal split. That means that neither three of them together conditioned the same secondary tonal split, nor two of them separately conditioned two different tonal splits, not to mention three of them conditioning three tonal splits separately. This leads us to the third relevant typology below.

(5)     Lingnan languages' phonological (tonological) Areal Type 3: The secondary tonal split in the high register, if any, was conditioned by either one or two initial sets from the three non-1C sets, aspirated (1A), unaspirated (1U), or glottalized (1G), limited to the following patterns: (a) 1A alone; (b) 1G alone; (c) 1A and 1G together; and (d) 1U and 1G together. That is, the secondary tonal split was never conditioned by 1U alone or 1A and 1U together.

It is not difficult to conceive of the reasoning behind (5). First, aspirated initials and glottalized initials have led to the result of depressing the pitch of the syllable in a number of languages (J. R. Zhang 1980, p. 38; Liao 2016b, pp. 170–71). Based on Southern Kam's three-way tonal split pattern in which the tones conditioned by aspirated initials have a lower pitch than the ones conditioned by unaspirated voiceless sounds, Edmondson (1990, p. 188) suggested that aspirations are not the direct cause of pitch lowering (as they, on the contrary, generally lead to an increase in F0), but rather, breathiness has caused this result in the process of breathy voice > aspiration > deaspiration, as breathiness is the typical pitch depressor. In this regard, Zhu et al. (2016, p. 18) pointed out that, while Edmondson is correct that aspiration is not the direct cause of pitch lowering, the sequence of breathy voice > aspiration > deaspiration would result in the initials becoming unaspirated after pitch lowering, but such a historical phonological change is contrary to the fact that Southern Kam's aspirated consonants are paired with lower pitch tones, not to mention that Edmondson's original breathy voice was voiceless (upper register) or voiced (lower register) in the history of the Kam language. They further argued that the breathy vowels led to the lower pitch, which would have gone through the process of initials aspiration > subsequent vowel invading the aspirated segment > breathiness with vowels > lower pitch. In addition, glottalized initials often bring about creakiness, an incidental feature of low pitch in many languages (Zhu and Yang 2010; Mai 2011, pp. 22–23). Therefore, it is understandable that both the aspirated sounds (1A) and the glottalized sounds (1G), alone or together, determine secondary pitch-lowering splitting tones.

Second, aspirated sounds (1A) and unaspirated stops (1U) are diametrically opposed in terms of laryngeal features, the former having a [+spread] feature and the latter a [-spread] feature, as aspiration tends to result in low tone features; the implication is that the unaspirated sounds (1U) do not follow the aspirated sounds (1A) in conditioning this secondary tonal split, and this point is resolved. Furthermore, although unaspirated stops (1U) do not cause secondary tonal splits alone, they share (+voiceless, -continuant, -aspirated) with the segment (ʔ-) in glottalized initials, which was indeed the condition of register voicing tonal split (Liao 2016b, p. 137). Therefore, in a number of dialects (e.g., Southern Kam, Standard Thai, and Baoxu Zhuang demonstrated above), in which secondary tonal splits were conditioned by glottalized initials (1G), there is the result of 1U later following 1G in conditioning secondary tonal splits.

In short, as 1A could not have conditioned secondary tonal splits together with 1U, the possibility that 1A, 1U, and 1G jointly conditioned secondary tonal splits does not exist, and, at most, only two sets, 1A and 1G, or 1G and 1U, could have conditioned secondary tonal splits.

*3.2. Lower Register Secondary Tonal Split as a Sinitic Origin Trait*

In the previous subsection, we focused our discussion on the secondary split of upper-register tones and confirmed that it is a trait propagated from the Kam-Tai languages to the Sinitic languages. In contrast, a secondary tonal split in the lower register has long

been found in Sinitic languages and has been much explored, but the status of its diffusion as a trait into the Tai languages seems never to have been explored. With the help of the framework of the tone-box chart, we discovered a secondary tonal split pattern that emerged in Sinitic and diffused into Kam-Tai (or at least Tai), i.e., a special secondary tonal split in the low/lower register, called *zhúoshǎng biàn qù* (濁上變去) or *zhuóshǎng guī qù* (濁上歸去) in traditional Chinese phonology. This is illustrated in the tone-box charts for the four Cantonese and Putonghua varieties in Tables 13–16 in Section 3.1 above, and is elaborated below.

In Table 13, the most distinctive feature of the Cantonese tones is Box B2O, where each of the three lexical items (i.e., three Chinese characters) has two tones. In other words, for the three characters 坐 'to sit', 斷 'to be broken', and 近 'near' in Box B2O (the *shǎng* tone conditioned by historical voiced obstruents), the tone of their literary reading is a low rising tone [23], in line with Box B2S (the *shǎng* tone conditioned by historical voiced sonorants), but the tone of their colloquial reading is a low flat tone [22], in line with Boxes C2S and C2O (the *qù* tone conditioned by historical voiced sounds). That is, if colloquial reading and literary reading were separated, Cantonese would have two different tone-box charts, and in the case of the chart reflecting the older level of colloquial reading, Box B2P would be the same color as Box B2S. i.e., the original tone B2 (lower *shǎng*) does not produce a secondary tonal split according to different laryngeal features of initials, but in the case of a chart reflecting a later level of literary reading, Box B2O would be a secondary tonal split from the original tone B2 (lower *shǎng*) and merge into tone C2 (Boxes C2S and C2O = lower *qù*), which is known in the history of Chinese phonology as the aforementioned *zhuóshǎng guī qù* 濁上歸去 or *zhuóshǎng biàn qù* 濁上變去, meaning "the *shǎng* tone conditioned by a voiced obstruent initial would be merged into the counterpart lower *qù* tone", as very commonly found in the direct descendants of LMC, such as Mandarin, Gan, and most Xiang Chinese varieties. Traces of the *zhuó shǎng guī qù* phenomenon can be found in the records from the 9th to 10th centuries AD in the Tang dynasty, and the first one that explicitly treats all characters with a voiced obstruent initial and the *shǎng* tone as homophones with the counterpart characters having the *qù* tone, as they are arranged together in *Zhou Deqing*'s (周德清) *Zhōngyuán Yīnyùn* (中原音韻) in the Yuan Dynasty (Tang 2013, p. 130). In contrast, in some other Sinitic languages, such as Yue, Wu, Min, and Hakka, at least in the colloquial reading, which reflects the older level, i.e., the pre-LMC level, these words are not incorporated into the counterpart *qù* tone, just as in the Cantonese case of colloquial readings for these items in Box B2O of Table 13. For these languages, the literary reading, which was systematically influenced by the phonology of the authoritative Chinese variety in the political center of Northern China in history from the period of LMC, has completed the process of *zhuó shǎng guī qù* as in Mandarin, and also as in the case of Cantonese literary readings in Box B2O in Table 13.

As for the situation of Binyang Luxu Pinghua in Table 14, the tones of Boxes B2O and B2S are consistent, without such a secondary split of Box B2O from B2, i.e., it appears that this dialect does not have a secondary tonal split in the lower register, but this is, in fact, only an appearance. If we increase the number of eligible Chinese characters in Box B2O, we will find that this dialect also has a similar secondary tonal split pattern to the literary reading of the characters in Box B2O in Cantonese, where the original tone *B (*shǎng*) potentially shifts to the counterpart *C (*qù*) tone if the original initials are voiced obstruents. However, as the differences between the two dialects are presented in Tables 13 and 14, there is no agreement regarding which characters in box C2O (with original voiced obstruent initials and a tone *B or tone *shǎng)* are shifted to the counterpart tone *qù* (*B) or not. Such inconsistency was also pointed out by de Sousa (Forthcoming) in his comparison of Nanning Weizilu Pinghua with Cantonese. Note that, in Weizilu Pinghua and Rongxian Yue, as shown in Tables 15 and 16, box B2O is also split into two tones, one of which is expectedly merged into the counterpart C2 tone, presenting the situation of *zhuó shǎng guī qù* in these two dialects.

If one says that the lower-register secondary tonal split in the four Sinitic language varieties noted above occurs only in the lower tone *shǎng* (B2), and this split is determined by the influence of Northern authoritative dialects in history, rather than by the laryngeal features of their own initial consonant system, we can look at the specific patterns of Weizilu Pinghua in Table 15 to give more evidence. We find that, in addition to *zhuóshǎng guī qù* in the *shǎng* lower tone (B2), the lower-register secondary tonal split also exists in checked syllables; that is, a split occurs between DL2S/DS2S and DL2O/DS2O, i.e., a secondary tonal split on the lower tone *rù* conditioned by original voiced obstruent initials. Furthermore, we also examine the tonal split patterns of some Hakka dialects, such as that of Hong Kong Hakka summarized in Table 17, to learn that the secondary tonal split in the lower-register tones is widespread in Sinitic languages.

**Table 17.** Tonal split patterns in Hong Kong Hakka.

|      | A   | B   | C              | DL | DS  |
|------|-----|-----|----------------|----|-----|
| 1A   | 223 | 331 | 53 (sandhi 55) | 33 | <u>33</u> |
| 1U   | 223 | 331 | 53 (sandhi 55) | 33 | <u>33</u> |
| 2S   | 21  | 331 | 53 (sandhi 55) | 33 | 33  |
|      |     | 223 |                | 55 | 55  |
| 2O   | 21  | 233 | 53 (sandhi 55) | 55 | 55  |

As shown, in Hong Kong Hakka, we first learn that the secondary tonal split occurs in the tones of lower *shǎng* (B2) and lower *qù* (D2), i.e., tone B2O, which was conditioned by the original voiced obstruent, split from B2 and merged into the *píng* upper tone (A1), while the checked syllable counterpart D2O tone also split from D2, becoming the complementary distribution allotone of *qù* on the checked syllables. Note that there are no vowel-length contrasts in Hakka, so tones *DL and *DS on checked syllables can be considered as a single *D. Second, we also see that a group of words (characters) with the *shǎng* lower tone (B2S) conditioned by the original voiced sonorants, mainly in colloquial readings, also follow the words with tone B2O to merge their original tone B2S into tone upper *píng* (A1), while the rest of the words, mainly of literary reading, have merged their tone into upper *shǎng* (B1), causing box B2S to have an internal split. This pattern is also seen in box D2S, where part of the words follow those with tone D2O to merge their tone into tone C, while the remaining words merge their tone into tone upper D. It can be seen that, although Hong Kong Hakka has only four tones, its tonal split/merger is quite complex, and this is reflected in the lower-register tonal split.

The above five examples of Sinitic languages fully illustrate that the lower-register secondary tonal split represented by *zhuóshǎng guī qù* is quite common in Sinitic languages. When we consider the *zhuóshǎng guī qù* phenomenon in the literary readings of the Southern Sinitic languages as a trait imported from the Northern authoritative dialects through borrowing, we cannot avoid examining the diffusion of this Sinitic trait into the Kam-Tai languages too. Indeed, this trait has actually diffused into the Kam-Tai languages through the Sinitic loanword system—see *zhuóshǎng guī qù* among the lexical items, with tone C2 represented by Yang Zhuang summarized in Appendix A, in which we can illustrate this by comparing some Sinitic languages and Yang Zhuang's MC loanword system. Table 18 shows a comparison of *zhuóshǎng guī qùi* in Lingnan languages, including Yang Zhuang, three Pinghua varieties and Cantonese, and Mandarin, which I have compiled and calculated from Appendix A, a survey of the development of the Chinese characters with *shǎng* tone conditioned by a MC voiced obstruent initial in some modern Lingnan languages and Putonghua.

**Table 18.** A comparison of the Chinese characters of *zhuóshǎng guī qù* among some modern Sinitic languages.

| | 156 Items (Chinese Characters) in Total | | | | | | | |
|---|---|---|---|---|---|---|---|---|
| | Eliminated Items | | Only One Reading | | | Two Readings (B2/C2, B2/-, -/C2; or -/C1) | % of Tones B2 | % of Tones C2 and C2 + C1 |
| | Missing | Irregular | B2 Tone | C2 Tone | C1 Tone | | | |
| Luxu PH | 13 | 18 | 63 | 49 | 13 | 0 | 40% | 31% and 40% |
| Tingzi PH | 13 | 23 | 30 | 81 | 9 | 0 | 19% | 52% and 81% |
| Weizulu PH | 9 | 23 | 24 | 77 | 22 | -/1 | 15% | 50% and 64% |
| Yang Zhuang MC loan | 11 | 8 | 5 | 127 | 4 | -/1 | 3% | 82% and 85% |
| Hong Kong Cantonese | 0 | 4 | 34 | 95 | 4 | 16/16; 1/-; -/1; -/1 | 33% | 72% and 75% |
| Guiliu (Liuzhou) | 0 | 0 | 12 | 144 | | 0 | 8% | 92% |
| Putonghua | 0 | 0 | 11 | 145 | | 0 | 7% | 93% |

The data in Table 18, taken from Appendix A, contain a total of 156 common Chinese characters in EMC whose initial consonant was a voiced obstruent and whose tone was the original *shǎng* tone (*B, corresponding to Tai *C) before the tonal split. As mentioned above, in accordance with the tone rules in the Chinese language, these items would either have developed to have a B2 tone (i.e., lower-register *shǎng* tone) after the tonal split in the descendant languages of MC (and in the LMC loanword system borrowed into Tai), or they would have developed to have a tone that has been merged into its counterpart C tone, i.e., tone C2 or the lower-register *qù* tone, as the result of the secondary tonal split in the low register of tone C. In the Mandarin dialects, the *zhuóshǎng guī qù* progression was completed so thoroughly that the 156 Chinese characters are now overwhelmingly pronounced in the *qù* tone, with 92% in Guiliu and 93% in Putonghua (Standard Mandarin), and only a few characters retaining their original *shǎng* tone. However, because the upper- and lower-register *shǎng* tones have been secondarily merged into a single *shǎng* tone (the modern tone B), and the upper- and lower-register *qù* tones have also merged into another single *qù* tone (the modern tone C) in the vast majority of Mandarin dialects, such as Guiliu and Putonghua, it is not possible to tell from the data of the Mandarin dialects alone whether these words had originally voiced obstruent initials; rather, it is necessary to rely on the lower-register *shǎng* or the lower-register *qù* tones of the southern Sinitic languages, such as Cantonese and Pinghua, to support this. Nevertheless, perhaps it was also the influence of Mandarin's predecessor, the authoritative Sinitic language variety during the LMC period in Northern China, that led the southern Sinitic languages to produce literary readings that are different from their inherited colloquial readings. This is why some of the 156 characters in these dialects (such as Pinghua varieties and Cantonese in Appendix A) of *zhuóshǎng guī qù* have also been misregistered by turning their tone into tone C1 (high register), instead of the expected tone C2 (low register), because the authoritative variety had previously merged the upper and lower *qù*. From this point, the *zhuóshǎng guī qù* or the secondary tonal split leading the merger from the LMC lower *shǎng* (B2) to the lower *qù* (C2) was not an independent development, but rather a trait imported from the Northern authority Sinitic variety in history. In any case, in our statistics, in addition to the lack of data for some items, if there is an aberrant change in tone, i.e., not C2, B2, or B1, such items are excluded from the comparison; therefore, for the comparison of the proportions, three entries are listed, i.e., the proportion of characters with tone C2 (or C, in the case of Mandarin), the proportion of characters with tone B2 (or B, in the case of Mandarin), and the proportion of characters pronounced with B2 plus B1 (which, although it wrongly reg-

isters as original initial voicing, it correctly registers as the tonal category as the result of *zhuóshǎng guī qù*).

As Table 18 shows, of the many languages in Lingnan, with the exception of Guiliu, which was the latest to come to Guangxi, where *zhuóshǎng guī qù* is consistently and thoroughly implemented as in other Mandarin dialects north of Lingnan, the rest of the languages are more or less conservative, i.e., the incidence of *zhuóshǎng guī qù* is lower than that in Mandarin dialects. Among these Lingnan non-Mandarin languages, Binyang Luxu Pinghua has the lowest number of words for *zhuóshǎng guī qù*, and, after eliminating 20% of the invalid data, 40% of the remaining 80% of the valid data, i.e., half of the words, are still read in the lower-register *shǎng* tone (B2); only 31% of the words are read in the lower-register *qù* tone (C2), and even with the deviations in the upper register *qù* tone (C1), the incidence of *zhuóshǎng guī qù* is still 40%, the same as the number of words still read in the lower-register *shǎng* tone (B2). In the other two Pinghua dialects (Nanning Tingzi and Nanning Weizilu), Hong Kong Cantonese, and Debao Urban Yang Zhuang, these words are more often pronounced in the lower-register *qù* tone (C2)[10]. In the above Pinghua dialects, if a word has completed the process of *zhuóshǎng guī qù*, then its original reading (which is supposed to be read in the original B2 tone) has usually been replaced, which means that there is mostly no co-existence of colloquial and literary readings, even though, sometimes, there are sporadic items that keep both readings, as demonstrated in Nanning Wezilu Pinghua by de Sousa (Forthcoming). Cantonese, however, has a significant number of words with different readings, so the proportions of colloquial and literary readings in Cantonese overlap, to some extent. Surprisingly, among the MC loanwords in Yang Zhuang, a non-Chinese language, *zhuóshǎng guī qù* occurs quite thoroughly after excluding 12% of invalid data, with only 3% of the remaining 88% of the words still being read in the lower-register *shǎng* tone and 82% in the lower-register *qù* tone; if we add the words read in the upper-register *qù* tone, the proportion of *zhuóshǎng guī qù* is actually as high as 85% within the valid 88%.

That is to say, diachronically speaking, in terms of tonal behavior, Binyang Luxu Pinghua is the most conservative of the many dialects showing Lingnan's *zhuóshǎng guī qù* trait, or, in other words, it is historically accepted to show a lesser degree of *zhuóshǎng guī qù* than the authoritative dialects of northern Sinitic, while the MC loanword system in Debao Yang Zhuang accepts it to the highest degree. This phenomenon is intriguing, because it reveals that the beginning of *zhuóshǎng guī qù* in the history of the Chinese language may have been earlier than thought. Note that, as in other Zhuang dialects, the Chinese loanwords in Yang Zhuang are borrowed in several historical layers, the oldest being from OC, followed by the Late Han, Early and LMC, Guiliu, Yue, and Putonghua layers; here, the LMC loanwords are those considered to have been borrowed from ancient Pinghua in the various Zhuang dialects (including the various languages of Central Tai and Northern Tai), around the time of the *Yùnjìng* 韻鏡 (c. 1150 AD), the oldest of the so-called rime tables, in the Song dynasty, and differs from the LM$^C$ (7th–11th centuries CE) that Pittayaporn (2014) adopted from Pulleyblank (1991) among the four strata defining Chinese loanwords in Southwestern Tai. In our previous discussion of the history of the development of Chinese tones, we also mentioned that the primary tonal register split occurred as late as the end of the Tang Dynasty (c. 10th century), which, in the present paper, is an important indicator of LMC's distinction from EMC. In the early Northern Song dynasty (11th century), after the Song court had suppressed the rebellion of the Zhuang leader *Nóng Zhìgāo* (儂智高), the northern troops sent to Guangxi brought with them a new LMC variety, which was also the precursor of Pinghua (de Sousa 2015a), so the early period of the Song dynasty can be seen as the beginning of the development of Pinghua in Guangxi. The loanwords borrowed from Pinghua after that period are the main body of MC loanwords that exist in the Zhuang dialects of Guangxi, and this Pinghua system of LMC loanwords is largely absent from the Southwestern Tai languages, echoing Pittayporn's use of the latest Chinese loanword layer to determine that the Southwestern Tai groups migrated out of Guangxi into the MSEA hinterland no earlier than Late Tang, and also indicating that

Southwestern Tai's migration should have been no later than the time of the *Nóng Zhìgāo* rebellion. Therefore, in Appendix A, I regard the loanwords borrowed from ancient Pinghua into Yang Zhuang as another LMC layer, while the OC, Late Han, EMC, and LMC in Pulleyblank (1991) and Pittayaporn (2014) shared by Northern Tai, Central Tai, and Southwestern Tai are collectively referred to as pre-Southwestern Tai (pre-SWT) Sinitic loanwords (these layers are directly identified in this paper using Pittayaporn's (2014) chronology). The following is the basis for the dating of *zhuóshǎng guī qù* from the evidence for pre-SWT Sinitic loans and LMC loans in Tai, represented by Yang Zhuang.

First, of the 137 entries in Yang Zhuang that fit the pattern of tone development, there are only four with lower *shǎng* (B2) and one with lower short *rù* (DS2) on checked syllables, but none of them are LMC loanwords borrowed from the ancient Pinghua system; rather, they are all pre-Southwestern Tai (pre-SWT) loanwords. These entries are $poj^{\prime 13\Omega\prime}$ 婢 'maid, daughter-in-law', $taw^{213\Omega}$ 杖 'crutch', $t\varepsilon a:\eta^{213\Omega}$ 象 'elephant', $t\varepsilon y:^{213\Omega}$ 社 'land's god', and $kjok^{21}$ 臼 'mortar'; the checked syllable in $kjok^{21}$ 臼 'mortar' suggests that it may have been borrowed from OC into Tai, vs. *C.[g]ʷəʔ, reconstructed by Baxter and Sagart (2014a). DS2 is also seen in the history of Tai tonal development as the checked-syllable counterpart of the lower *shǎng* (Sinitic B2 or Tai's C2); therefore, it is also considered as conforming to the rule. These etyma are also commonly shared among the Southwestern Tai languages, e.g., the five etyma above in Thai are $p^haj^{453\Omega}$ 'daughter-in-law', $t^ha:w^{453\Omega}$ 'crutch', $t\varepsilon^ha:\eta^{453\Omega}$ 'elephant', $t\varepsilon^h\ w\partial^{453\Omega}$ 'lineage', and $k^hrok^{45}$ 'mortar', indicating that the most tonally conservative words are all borrowed from Chinese earlier than the EMC period.

Second, of the rest, 132 words are read in *qù* tones, including 128 in the lower register *qù* and 4 in the upper register *qù*, which is evidence that Yang Zhuang's MC loanwords are quite thorough in *zhuóshǎng guī qù*. Of the 128 characters reading in the lower *qù* tone, 7 belong to the pre-SWT Sinitic loanwords, such as et $te:\eta^{33}$ 錠 'strip' (vs. Thai $t^h\varepsilon:\eta^{41\Omega}$), $te:m^{33}$ 簟 'bamboo mat' (vs. Tai Lue $t\varepsilon:m^{33}$), and $po:^{33}$ 父 'father; male' (vs. Thai $p^ho:^{41\Omega}$), which are mostly found in Southwestern Tai and are also with the lower *qù* tone; in addition, two of the four upper-register words may also be pre-SWT loanwords. The fact that these characters belong to the pre-SWT loanword system are read in tone *qù*, rather than the original tone *shǎng*, suggests that the phenomenon of *zhuóshǎng guī qù* predates the migration of Southwestern Tai from its ancestral homeland of Guangxi, which occurred before the end of the Tang Dynasty around 1000 years ago. In this way, these items were actually borrowed into Tai before the register tonal split of Tai. It was only after the register tonal split in Tai that the tone of these words developed into a lower-register *qù*, due to the decisive effect of the voicing of the consonants. Such a chronological order is simply displayed in Figure 5.

Note that the Tai languages did not have register tonal splits until after the Chinese language had completed its register tonal split, and the fact that the Thai writing created around 700 years ago has only two tone markers to distinguish the four proto tonal categories, namely tones *A and *D with no marker, and *B and *C with two markers (Brown 1985), shows that the register tonal split occurred much later than that in Chinese. In this case, it is only reasonable to consider the *zhuóshǎng guī qù* phenomenon as having occurred before the latest limit of the primary tonal split, i.e., before the end of the Tang dynasty.

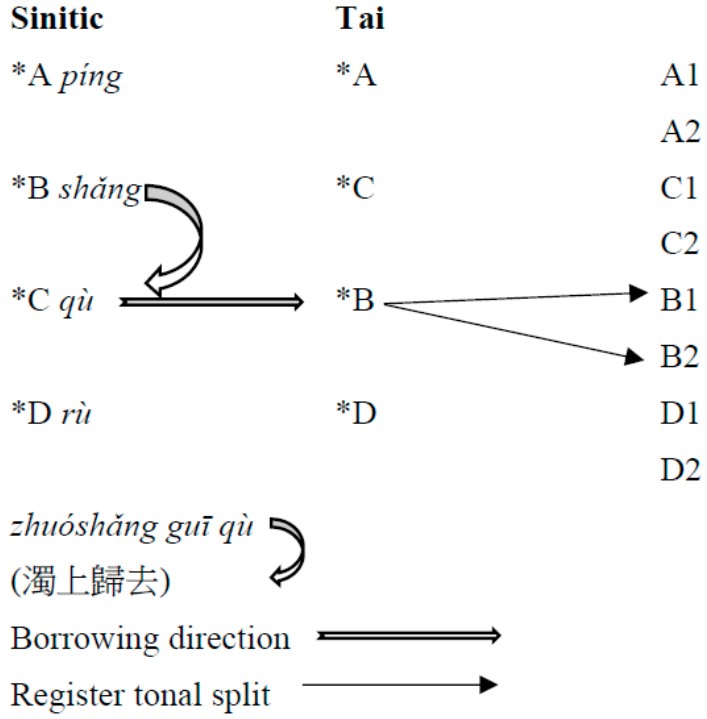

**Figure 5.** Diachrony of *zhuóshàng guī qù* of Middle Chinese loanwords in Tai languages.

The preliminary conclusion we can draw from the above discussion is that this lower register secondary tonal split is embedded in Tai as a trait. However, unlike the upper-register secondary tonal split, which is automatically conditioned by the laryngeal features of the initial consonant, it is achieved through a group of loanwords. This group of loanwords, some of which were borrowed in the pre-SWT period, also directly influenced the remaining post-SWT loanwords in this group, i.e., the LMC loanwords borrowed through Pinghua, and the vast majority of these *zhuóshǎng* words have merged into the counterpart tone *qù*. Compared with Yang Zhuang's Middle Chinese loanwords, the proportion of *zhuóshǎng* words incorporated into the counterpart tone *qù* is lower in modern Pinghua dialects, especially those represented by Hengxian Luxu. The reason for this is most likely that Pinghua, as a Sinitic language, had its own developmental trajectory; it might not have experienced *zhuóshǎng guī qù* itself before it moved into Guangxi, and it was only after it moved into Guangxi that *zhuóshǎng guī qù* gradually occurred under the influence of the authoritative dialects of the north, but it still retained more of its inherited lower *shǎng* tone in its descendant varieties. In contrast, the Tai dialects received more Middle Chinese loanwords from Pinghua than from pre-SWT Sinitic layers, but because *zhuóshǎng guī qù* had already occurred from these previous Sinitic layers, the *zhuóshǎng* items were automatically assigned as having the counterpart *qù* tone, even in the post-SWT layers. Although *zhuóshǎng guī qù* in Chinese needs to be analyzed and examined in more depth in the light of the Tai corroboration to explain why it occurs so thoroughly in the Middle Chinese loanword system in Tai languages, we can at least conclude here that the lower-register secondary tonal split occurs in tone *C of Tai. Of course, strictly speaking, it can only be seen as a result of absorbing a group of Sinitic loanwords, not as a result of being conditioned by Tai's own initial laryngeal features, because, in Tai's inherited etyma, even if the proto-Tai initial was a voiced obstruent, it did not cause any secondary tonal split in the lower-register tone. Therefore, there is no way to add a horizontal line of laryngeal features to the lower register in the Tai tone-box chart[11].

In fact, if we ignore the genetic relationship between the southern Sinitic languages, Pinghua and Yue, and the direct descendants of the northern authoritative Sinitic dialects, represented by Mandarin, we can also consider the literary readings in these southern Sinitic languages as a layer of "loanwords" borrowed from the historically authoritative Sinitic language after the EMC period. That is, the apparent tonal split of the items in Box B2O is actually not a tonal split from one box, but is simply a tone belonging to the later historical levels of the same Chinese character borrowed from the authoritative language in history, based on the tonal correspondence between these southern Sinitic/Kam-Tai languages and the authoritative Sinitic languages. In this respect, the nature of the lower-register secondary tonal split as a Sinitic-origin trait is different from that of the upper-register secondary tonal split in the Kam-Tai languages. The former is transmitted through the tonal correspondence of the loanword system, whereas the latter is a pattern conditioned by the laryngeal features of a language's own initial consonants. However, we still regard the tone of the etyma involved in *zhuóshǎng guī qù* as a tonal split on the original lower-register tone B2 for two reasons. One, it is indeed a laryngeal feature (original voiced obstruents) to cause such a split for the direct descendants of MC, namely Mandarin, Xiang, and Gan. Secondly, for dialects such as Cantonese, Pinghua, and Hakka, the sound rule of *zhuóshǎng guī qù* has been established within their own sound system because of the strong influence of the written language.

Finally, in this subsection, we summarize the relevant Lingnan areal trait and areal type as in (6) and (7), respectively.

(6)     Areal Trait 3 of the tonal behavior in Lingnan languages: The pattern of the secondary tonal split of the lower-register tones spreads among different subgroups of Sinitic languages and has diffused into some Kam-Tai languages, mainly limited to be in the Middle Chinese loanword system.

(7)     Lingnan languages' phonological (tonological) Areal Type 4: The lower-register tonal split pattern is limited to two secondary registers, because in both Sinitic and Kam-Tai, only two laryngeal features are found in the original voiced initials, namely voiced sonorants and obstruents.

### 3.3. Upper-Register Tones Associated with Sonorant Initials as of a Kam-Tai Origin Trait

In this subsection, another tonological trait that originates in Kam-Tai also has infiltrated some Lingnan Sinitic languages through loanwords or substratum. To elaborate on this, a further interpretation of the diachrony of the tonal split is needed, as it is crucial for determining this trait as originating from the Kam-Tai languages. We already referred to the primary and secondary tonal split patterns of the Kam-Tai languages above, and mentioned that "primary" and "secondary" are defined in terms of diachrony for reasons that were suggested in some previous studies, namely Li (1977, pp. 25–17), Zhang (1980), Liang and Zhang (1996, pp. 816–17), Zhang et al. (1999, pp. 243–46), and Liao (2016a, pp. 84–86; 2016b, pp. 121–30). The following discussion of the diachrony of tonal split is largely extracted from these studies.

Unlike Chinese, where the register tonal split was caused by the devoicing of original voiced obstruent initials (2O), the register tonal split in Kam-Tai languages was most likely caused by the voicing of original voiceless (simultaneously pre-aspirated) continuant (or sonorant) initials (1C). That is, for the Sinitic languages, after the devoicing of the initial set 1O triggered the register upper–lower tonal split, the set of 1S (original voiced sonorants) automatically assigned its conditioning tones to the lower-register tone to contrast with the upper-register tone conditioned by the original voiceless obstruents (both aspirated and unaspirated, i.e., 1A and 1U). Meanwhile, for the Kam-Tai languages, after the voicing of initial set 1C caused the register tonal split, the other voiceless initial sets (1U, 1A, and 1G) also automatically assigned their conditioning tones to the upper-register tone to contrast with the lower-register tone conditioned by the original voiced sonorant/obstruent initials. These are the processes known as the primary tonal split in Sinitic and Kam-Tai languages. Later, either the further tonal split that commonly occurred in the upper-register tones in

the Kam-Tai languages or the further tonal splits that are prevalent in the lower-register tones in the Sinitic languages, as well as the further splits that occurred in the checked syllables in Kam-Tai languages and in some Lingnan Sinitic languages according to vowel length, are all regarded as secondary tonal splits; i.e., they are more recent in chronological order.

The evidence for these inferences is largely based on the fact that the original voiced obstruents of certain modern Tai dialects have not yet completed the devoicing process, such as Cao Bang Tho of Central Tai (Haudricourt 1972, p. 65), Dai Zhuang of Central Tai (i.e., Wenma Zhuang of Southern Zhuang) (L-Thongkum 1997, p. 22), Yizhou Suogan of Northern Zhuang, and Baiji of Yongnan Zhuang (Liao 2016b, p. 126), but these dialects have long completed the register tonal split caused by the voicing of original voiceless continuants. In addition, Thai orthography, which was formulated around 700 years ago, still uses original voiced initial letters to write the lower-register voiceless obstruent initials, reflecting the fact that the devoicing of Thai original voiced obstruents was much later than the primary tonal split (L-Thongkum 1997, pp. 207–8; Liao 2016b, pp. 125–26). For the typological plausibility of sound change, L-Thongkum (1997, p. 209) cites Ladefoged's (1971, p. 11) point that voiceless sonorants are actually "partially voiced" before vowels, further pointing out that it takes longer to go from *$m$- (partially voiced) to $m$-, i.e., *$\underset{\circ}{m}$-[$m^m$] > $m$-, than to go from *b- (fully voiced) to $p$-, i.e., *b- > *$\underset{\circ}{b}$- > $p$-, or from *b- (fully voiced) to $p^h$-, i.e., *b- > *$\underset{\circ}{b}$- > *bh- > $p^h$-. Therefore, it would make sense that the tonal split in the Kam-Tai languages was caused by the voicing of voiceless sonorants, rather than the devoicing of voiced obstruents. Furthermore, L-Thongkum (1997, pp. 209–12) pointed out that the historically voiceless sonorants also played an important role of triggering the tonal split in the Mon-Khmer, Hmong-Mien, and Tibeto-Burman languages, causing pitch raising that led to the tonal split, so this phenomenon is cross-linguistic. For Kam-Tai languages, the decisive role that the voicing of voiceless sonorants played in the history of the primary tonal split is not an isolated phenomenon.

As for the definition of the secondary tonal split in the Kam-Tai languages as being later than the primary tonal split in the chronological evolution, Liao (2016a, pp. 86–89; 2016b, pp. 131–39) summarized some of the previous studies and added other evidence of his own to draw inferences. The following two points are what I consider to be the most important. One is that, although non-straightforward secondary tonal split patterns can be found in all terminal subgroups of the Kam-Tai languages, all of these subgroups, except Saek, also have dialectal varieties that still account for more than half of the modern Kam-Sui languages that retain the straightforward primary pattern. For example, in Yang Zhuang, there are Jingxi and Napo varieties that maintain the primary tonal split patterns and Debao varieties that have developed complex secondary tonal split patterns. The only way to explain this is that all the varieties started out as having primary tonal split patterns, and then the Debao varieties themselves underwent a secondary tonal split. Otherwise, it is impossible to explain why the Jingxi and Napo varieties, which are more closely related to the Debao varieties, differ from the Debao varieties in their straightforward tonal split, and instead have such consistent straightforward patterns as almost half of the varieties from the other Kam-Tam subgroups, such as Tai Lue (Southwestern Tai), Wuming Zhuang (Northern Zhuang), Shuangding Zhuang (Yongnan Zhuang), and Sui (Kam-Sui). Second, as with the secondary tonal split patterns summarized in Section 3.1, in dialects where a secondary tonal split occurred, further tonal splits according to the laryngeal features of the initial sets on upper-register tones are limited to four possible cases, namely set 1A alone, set 1G alone, sets 1A and 1G together, and sets 1G and 1U together, to condition the possible secondary tonal split. The tone conditioned by set 1C, which is treated as a result of the primary tonal split, is always in opposition to the lower-register tones (except in a few dialects where the upper and lower-register tones have been reunited). It is also important to note that secondary tonal splits rarely occur neatly in all five proto-categories as they do in Southern Kam (in fact, the only Kam-Tai language known to have such a pattern is Southern Kam). More dialects where a secondary tonal split occurs are restricted

to one tonal category (e.g., Thai, Northern Thai, and Hurun Yang Zhuang), two tonal categories (e.g., Tiandong Zhuang), or three tonal categories (e.g., Debao Yang Zhuang), while the other tone categories maintain the primary straightforward pattern, which further exemplifies the irregularity of the secondary tonal split. All these findings suggest that the secondary tonal split exists as a diffused trait in the various terminal subgroups of the Kam-Tai languages and could not have been present at the outset, hence creating such inconsistent patterns spread out in different Kam-Tai subgroups.

Furthermore, the dialects in which the secondary tonal split occurs, such as Southern Kam, Xialeng Zhuang, and Baoxu Zhuang, which we introduced above in Section 3.1, are those in which one or all proto-tones underwent a three-way split, i.e., one or all of the upper-register tones split into two tones, in opposition to the lower-register tone(s). However, as in more dialects where a secondary tonal split occurs, such as Thai and Debao Urban Yang Zhuang illustrated in Section 2, the upper splitting tones merged into the lower-register tones, forming a new, non-straightforward two-way split. In particular, the Thai pattern shown in Table 4 synchronically presents a primary straightforward two-way split between historical voiceless sounds (1F, 1U, and 1G) and voiced sounds (2) on PT Tones *BCD, but shows a secondary two-way split as 1F-1U/1G/2 on Tone *A. However, this has been proven to be a result of the later merger of the splitting tone A1U/G and the lower-register tone A2, i.e., from a three-way split to a new two-way split, in recent history after the 17th century (Pittayaporn 2018; Tingsabadh 2001). Liao (2016a, 2016b) also offered his opinion on this issue based on some dialectal evidence, specifically the Huashan Min Zhuang patterns, that all dialects with secondary non-straightforward two-way splits first underwent a three-way split like Southern Kam, but the limitations of the tonal load carried by that language caused the secondary splitting tone to be subsumed into the lower-register tone to form a new two-way split pattern. Refer to Liao (2016a, pp. 88–89; 2016b, pp. 168–72) for the details. In any case, the most important point here is that, regardless of whether a secondary splitting tone is kept as a distinctive tone or has been merged into the lower-register tone, it must have not been conditioned by the original voiceless continuants (sonorants, 1C), but by the other three groups (1A, 1G, and 1U).

In addition, there are a number of dialects in Zhuang that have had long-standing contact with Chinese, especially urban varieties, such as the varieties spoken in the Jingxi (靖西), Napo (那坡), Daxin (大新), Tiandeng (天等), and Tianyang (田陽) county seats; in none of these, the original preglottalized initials *ʔb-, *ʔd-, *ʔj-, and *ʔw- are retained, but all of them are merged into the counterpart continuant initials /m-/, /n-/, /j-/, and /w-/, respectively. This merger must have occurred well after the primary tonal split so that, in most of these dialects, the original preglottalized initials merged with the high-register sonorants, or, in other words, although they have merged into the modern voiced sonorants, the tones conditioned by them in history were the high-register tones, the same as those conditioned by the original voiceless continuants. Unlike them, the Tiandeng urban variety clearly shows that this merger occurred after the secondary tonal split, because, in this dialect, tone *A had a secondary tonal split conditioned by the original glottalized initials, and the tone value of this splitting tone A1G merged into the lower-register tone A2 to complete a new non-straightforward two-way split. Only if the secondary tonal split had occurred first, and the processes *ʔb- > m-, *ʔd- > n-, *ʔj- > j-, and *ʔw- > w- were completed after that, can it be reasonably explained why the tone value of A1G conditioned by these original preglottalized initials was the same as that of A2. The most important point to be made here is that pre-glottalized initials are another important source of the seemingly high-register sonorants in modern dialects where deglottalization has been completed. This is because some of the lexical items that combine a sonorant initial and an upper-register tone in the Lingnan Sinitic languages that we are going to discuss below are believed to have come from the Kam-Tai substratum or loanwords, and many of these sonorants were, in fact, historically preglottalized initials in the related forms in the Kam-Tai languages.

In contrast to the important role of the original voiceless continuant initials in the process of the register tonal split in the Kam-Tai languages, in the Sinitic languages, this role

has been taken up by the voiced obstruent initials. In our previous discussion of the designs of the integrated Kam-Tai tone-box chart and the integrated Lingnan Sinitic tone-box chart, we have emphasized that MC did not have the same voiceless sonorant and glottalized initials as the Kam-Tai languages by the time of the tonal split, so the trigger of the Chinese primary tonal split, which was limited to the devoicing of the obstruent initials, was also reasonable. Note that the so-called 'zero initial' (零聲母) is called the *yǐng* initial (影母) in traditional Chinese phonology and is generally reconstructed as a glottalized stop *ʔ-, and always conditioned tones together with original unaspirated voiceless obstruents in Sinitic languages. Thus, it is categorized as belonging to the 全清 (clear) or unaspirated surd (1U) set, which is different from Kam-Tai's glottalized initials (including *ʔ- and all preglottalized initials discussed above) in term of tonal split conditioning. As the original voiceless sonorant and glottalized initials did not exist in MC, the predecessor of the majority of the modern Sinitic languages, it is clear that, in some modern Lingnan Sinitic languages, such as Cantonese and Pinghua, lexical items consisting of sonorant consonants associated with the upper-register tones are more likely Kam-Tai substratum words or loanwords from Kam-Tai. This is somewhat similar to the way in which the lower-register tonal split was diffused from Sinitic to Kam-Tai languages through a certain number of loanwords, as demonstrated in Section 3.2.

Note that, in Sinitic languages, there are a number of Sinitic-origin etyma with modern sonorant initials associated with the upper-register tones, but these items either did not have a voiceless sonorant consonant in MC or underwent a later tonal change. For example, items with the bilabial approximant initial (which is phonetically sonorant or continuant) for the upper *píng* tone (A1) items in Chinese languages mostly historically had the *yǐng* initial (影母), which is generally reconstructed as a glottalized stop *ʔ-, such as the Cantonese /wa$^{55}$/ 蛙 'frog' and /wɐn$^{55}$/ 溫 'warm'. Their modern initial /w-/ is a development of deglottalization after the tonal split. For another example, the Mandarin *māo* and Cantonese /mau$^{55}$/ 貓 'cat', which had a *míng* initial (明母) and original *shǎng* tone (*B) in MC, underwent an aberrant tone change to the upper *píng* tone (A1) in most modern Chinese dialects. In addition, in Cantonese, there are some semi-muddy (次濁, Sonorant/Liquid) (2S) initials associated with the tone *rù* (*D) on checked syllables that merge into the upper-register tone, e.g., 粒 /nap$^{55}$/ (DS2S > DS1) 'grain; classifier of a small round object'. Another example is the Cantonese 剝 /mɔk$^{55}$/ 'to take off; to remove' with a sonorant and a high-register tone *rù* (DL1), but this item actually had a voiceless stop initial *p- (幫母) in MC; thus, the modern Cantonese initial /m-/ is an aberrant initial change after tonal split. Some modern English or foreign loanwords in Cantonese also carry a high-register tone associated with a sonorant initial, such as /mɐk$^{55}$/ from the English 'mark'. With such exceptions, in these Lingnan Sinitic languages, most of the remaining lexical items with sonorant initials associated with the upper-register tones are of non-Sinitic origin, and a large proportion is likely to be of Kam-Tai origin. This has already been pointed out by Li (1992, p. 331), who suggested that the presence of such a lexicon of sonorant initial and the upper-register tone combination in Cantonese may have been the result of an early contact relationship between the two groups of Yue Chinese and Tai languages. Using Cantonese as an example, we have collected some of the vocabulary in Table 19. Note that the Tai meanings are similar to those in Cantonese, unless otherwise stated.

In Table 19, some of the Chinese characters that we first see are dialectal characters of Cantonese (e.g., 睺, 搣, and 搵), indicating that they do not exist in other non-Lingnan Sinitic languages. Most of the other characters, although ordinary Chinese characters, have a different meaning in the Cantonese spoken language than in the written language, suggesting that they are used only to represent close or homophonic Cantonese colloquial words and have no etymological connection with the original etymology of the Chinese character. All of these words, in Cantonese, have a sonorant initial paired with an upper-register tone, suggesting that they may have been borrowed from the Kam-Tai languages into Cantonese. As we mentioned above, some of these words, such as 烚/nat$^{33}$/'hot', 擘/mak$^{33}$/'split', 乜/ 歪/mɛ$^{25}$/'askew', and 搣/mit$^{55}$/'pinch', originally had preglottalized

initials in the Kam-Tai languages, but such initials are not found in Yue Chinese, and the preglottalized initials are neutralized into the counterpart POA (place of articulation) sonorants when borrowed, as is the case with some urban Zhuang varieties.

**Table 19.** Some Cantonese colloquial words with a sonorant initial associated with an upper-register tone.

| Character | Reading and Tone | Meaning | Possible Related Tai Root: PT and Modern Tai Forms |
|---|---|---|---|
| 罅 | $la^{33}$ C1 | Gap | *$gle:^B$ > $ke:^{B2}$ Debao |
| 舐 | $laj^{25}$ B1 | Lick | *$li\partial^A$ > $li:^{A2}$ Debao, $li\partial^{A2}$ Thai, $la:j^{C2}$ Guangnan |
| 舔 | $l\varepsilon m^{25}$ B1 | Lick | *$gle:m^C$ > $kle:m^{C2}$ Hengxian |
| 冧/ 㷠 | $l\mathit{e}m^{33}$ C1 | Collapse | *$^h lom^B$ > $lam^{B1C}$ Debao, $lom^{B1C}$ Guangnan |
| 躝 | $lan^{55}$ A1 | Crawl | *$gla:n^A$ > $kja:n^{A2}$ Debao, $k^h la:n^{A2}$ Thai |
| 擸 | $lap^{33}$ DL1 | Take on (all) | *$^h na:p^{DL}$ > $na:p^{DL1C}$ 'to clamp' Debao |
| 凹 | $l\mathit{e}p^{55}$ DS1 | Concave | *$ʔbup^{DS}$ > $ʔbup^{DS1G}$ Debao, $bup^{DS1G}$ Thai |
| 甩 | $l\mathit{e}t^{55}$ DS1 | Drop off | *$^h lʊt^{DS}$ > $lut^{DS1C}$ Debao, $lut^{DS1C}$ Thai |
| 佬 | $low^{25}$ B1 | Fellow, guy | *$la:w^A$ > $la:w^{A2}$ 'Tai/Lao tribe' Thai, Lao |
| 嚦 | $l\varepsilon\eta^{55}$ C1 | Young (kids) | *$lu:k^{DL} ʔe:\eta^A$ > $luk^{DL2} ʔe:\eta^{A1G}$ Debao |
| 焫 | $nat^{33}$ DL1 | Hot | *$ʔd\mathit{ɯ}a:t^{DL}$ > $ʔdu:t^{DL1G}$ Debao, $d\mathit{ɯ}\partial t^{DL1G}$ Thai, $ʔda:t^{DL1G}$ Wuming |
| 諗 | $n\mathit{e}m^{25}$ B1 | Think | *$^h nam^C$ > $nam^{C1C}$ Debao, Wuming |
| 嬲 | $n\mathit{e}w^{55}$ A1 | Angry | *$^h naw^B$ > $naw^{B1C}$ Debao, Daxin, $naw^{A1C}$ Hengxian |
| 奀 | $\eta\mathit{e}n^{55}$ A1 | Thin/small | *$^h \mathit{ɲ}an^B$ > $\eta^j an^{B1C}$ Debao |
| 躟 | $\eta\mathit{e}n^{33}$ C1 | Vibrate | *$ɬan^B$ > $ɬan^{B1C}$ Debao, $san^{B1C}$ Thai |
| 呃 | $\eta ak^{55}$ DS1 | Cheat | *$^h lo:k^{DL}$ > $lo:k^{DL1C}$ Debao, $lɔ:k^{DL1C}$ Thai |
| 擘 | $mak^{33}$ DL1 | Split | *$ʔba:k^{DL}$ > $ʔba:k^{DL1G}$ Debao |
| 乜/ 歪 | $m\varepsilon^{25}$ B1 | Askew | *$ʔbi\partial w^C$ > $ʔbe:w^{C1G}$ Debao, Hengxian, $bi\partial w^{C1G}$ Thai |
| 搣 | $mit^{55}$ DL1 | Pinch | *$ʔbit^{DS}$ > $ʔb\partial t^{DS1G}$ Debao, $bit^{C1G}$ Thai |
| 孖 | $ma^{55}$ A1 | Twin | *$pwa:^A$ > $p^h a:^{A1A}$ Debao, $fa:^{A1F}$ Thai |
| 扔 | $wing^{55}$ A1 | Throw off | *$ɣwe:\eta^B$ > $we:\eta^{B2}$ Debao, $k^h wa:\eta^{B2}$ Tai Lue |
| 搵 | $w\mathit{e}n^{25}$ B1 | Seek, find | *$wan^B$ > $wan^{B2}$ 'to dig out' Debao |
| 郁 | $jʊk^{55}$ DS1 | Move | *$ʔjok^{DS}$ > $ʔjɔk^{DS1G}$ 'to poke' Debao |

In addition, there are words that do not have an upper-register tone in Kam-Tai languages, such as *$li\partial^A$ 'lick', *$gle:m^C$ 'lick', and *$gla:n^A$ 'crawl', all with voiced initials in PT, so they all have a lower-register tone in modern Tai dialects. However, Cantonese adapted them with an upper-register tone, suggesting that, at the time of borrowing into Cantonese, the tonal split had already ended, and that Cantonese did not apply the historical tonal correspondence to borrow them, but may have applied approximate tonal values. This may

have had something to do with the tonal flip-flop of some Tai languages we mentioned earlier. For example, in Longsang Zhuang, the original upper-register tones developed into low-pitch tones to shift the original lower-register tones into high-pitch tones so that the B1 and B2 tones in this dialect had values of mid-falling (31) and high-falling (53), respectively, forming a typical flip-flop. If the Cantonese /win⁵⁵/(tone A1) 'throw off' (in Table 19) was indeed borrowed from a Kam-Tai dialect that was historically similar to Longsang Zhuang in terms of tonal flip-flop, the situation is understandable. Even though the word has an original lower-register tone B2 in Kam-Tai, it is indeed /weːŋ⁵³/ (<*ɣweːŋB) in Longsang Zhuang, which has a high-pitch tone for it. Therefore, Cantonese probably borrowed it using its tone A1, a high-pitch tone (53 or 55).

In any case, the origin of a number of Cantonese substrate words is more likely to be Kam-Tai languages, where they are commonly found, while their distribution among Sinitic languages is limited, even within the Yue varieties.

The above examples in Table 19 show that sonorant initials associating with an upper-register tone are a trait that diffused from the Kam-Tai languages to the Lingnan Sinitic languages (mainly Yue and Pinghua) through loanwords or substratum, but this does not mean that all the words in the table are necessarily of Kam-Tai origin. Their origins are debatable. The origin of the word 諗 /nɐm²⁵/ 'to think', for example, is controversial. Bai (1980, p. 217), for instance, associates its etymology with the word 恁 in *Jiyun*, a rime dictionary of MC published in 1037 during the Song Dynasty. Wang (1999) constructed this word as *ȵiěm, while Li (1980) constructed it as *ńźjəm, but both failed in explaining the tonal behavior of 諗 /nɐm²⁵/ in Cantonese. Bauer (1996, pp. 1832–33) noted that, although the word is found in Cantonese, Zhuang, and Yao, Ouyang (1989, p. 611) suggested that it comes from Zhuang, etyma with the same or similar meaning as 'to think, consider' in the other Kra-Dai languages are etymologically inconsistent across dialects. Therefore, he considers it premature to define the origin of the Cantonese word /nam²⁵/ 'to think'. However, it is also certain that, even if the word did not enter Cantonese directly from the Tai languages, given that it is an item with a sonorant consonant with an upper-register tone in both Zhuang and Cantonese, it is very unlikely that it is a Sinitic etymon, but more likely first appeared in Zhuang languages and then entered Cantonese[12]. In short, in this subsection, we summarize one more tonal behavior as a Lingnan areal trait as in (8).

(8)    Areal Trait 4 of tonal behavior in Lingnan languages: The pattern of sonorant initials associating with an upper-register tone is a Kam-Tai origin trait, which has diffused into some Sinitic languages, mainly Yue and Pinghua.

## 4. Conclusions

After comparing the commonalities in the tonal behavior of the two most populous language groups in the core Lingnan area, Sinitic and Kam-Tai, some major findings are drawn. In addition to the secondary tonal split pattern on checked syllables which have been suggested to have diffused from Kam-Tai to neighboring Sinitic languages, particularly Cantonese and Pinghua, the secondary tonal split patterns of the upper-register tones as areal traits are also suggested to be of Kam-Tai origin and have diffused to these Lingnan Sinitic languages. In contrast, as the presentative lower-register tonal split pattern, *zhuóshǎng guī qù* was first processed in northern authoritative Sinitic languages and diffused to Lingnan languages, both Sinitic and Kam-Tai, in the form of a lexical category of loanwords. Further, the upper-register tone associated with sonorant initials in Sinitic languages in this area is also a Kam-Tai origin trait, or specifically part of the Kam-Tai substratum. Areal typological issues involved in these secondary tonal split patterns were also determined, focusing on different upper limits for possible secondary splits and the participation of particular sound sets.

During the exploration of the above findings, we are confident that the following methodological contributions to the field have been made in this study.

The first issue is the criteria and argumentation for and against inherited features vs. areal traits. As tonogenesis and tone as areal traits are more plausible explanations

for the performance of tones and for the correspondence of tones among related and unrelated languages, there is a growing consensus that tones are an emergent linguistic phenomenon, which forms one of the theoretical foundations of this paper. Thus, it is believed that the tones in all Kra-Dai languages belong to a trait that has been accepted from the earlier Sinitic languages. However, after a cross-family comparison of the tonal behavior of the Lingnan languages, this study is the first to divide the discussion of tonal traits into two chronological phases, i.e., the early one-way diffusion from Sinitic to Kra-Dai in the broader areas, and the later two-way interaction between Kam-Tai and Sinitic languages centered on Lingnan, with Kam-Tai to Sinitic traits dominating. In the case of the upper register tonal split patterns, for example, the original voiceless initial consonants of most Kam-Tai languages had aspirated stops, pre-aspirated continuants, unaspirated stops, and glottalized sounds by the time of the tonal split, two more sets of laryngeal features than the two sets of aspirated and unaspirated surds in the Sinitic originally voiceless initial consonants. Therefore, the upper register tonal splits conditioned by laryngeal features are naturally more diverse in the Kam-Tai languages than in the Sinitic languages, and the diffusion of these patterns from the Kam-Tai to the Sinitic languages through close contact in Lingnan is natural.

The second point Is the improvement in the inadequate use of tone-box theories in previous studies. These frameworks include an integrated Tai tone-box chart (Liao 2022) that can be applied to all Kam-Tai languages based on the Southwestern Tai tone-box chart designed by Gedney ([1972] 1989) half a century ago, and the similar Sinitic tone-box chart devised in this paper based on this theory. These frameworks make it possible to present, at a glance, the tonal correspondence between languages that are genetically related to each other and between languages that are not related to each other. In combination with the empirical data significantly increasing in recent decades, these frameworks can reveal new insights and cross-family correlations in a convenient sense.

Another point is that the tonal evidence in this study was used to propose a more fine-grained relative chronology of changes under contact between Sinitic and Kam-Tai, as well as criteria to detect Kam-Tai borrowings in Lingnan Sinitic languages. The chronology can be represented by the dating of the *zhuóshǎng guī qù* phenomenon. We can tell from the existence of *zhuóshǎng guī qù* found in some pre-SWT Sinitic loanwords that this phenomenon must have occurred in Sinitic languages before the 8th century, at the upper bound, when the SWT speakers migrated out of the Lingnan area. As Thai documentation attests to the fact that no primary tonal split occurred in Thai until 700 years ago, the borrowing of loanwords with *zhuóshǎng guī qù* into Kam-Tai precedes the occurrence of tonal splits. There are, of course, some unanswerable questions in this paper that need to be addressed in further studies. Although the complicated secondary tonal split patterns presented in this study are concentrated in Kam-Tai and some of the Sinitic languages in the core Lingnan area, and in the hinterland of MSEA along with the immigration of Southwestern Tai languages, it cannot be eliminated that such patterns spread as areal traits in other language families as well. For example, Haudricourt (1972) has long described similar two-way and three-way split patterns in some Hmong-Mien languages, but because of the paucity of data and the lack of more in-depth research, we do not know how such secondary tonal splits are distributed in these languages and how they relate to similar patterns in Kam-Tai and Sinitic. We believe that further study of tonal behavior among Hmong-Mien, Kam-Tai, and Sinitic can reveal more new insights and cross-family correlations. In our hypothesis, it was also the influence of Mandarin's predecessor, the authoritative Sinitic language variety during the LMC period in Northern China, that led the southern Sinitic languages to produce literary readings that are different from their inherited colloquial reading. This matter should also be further attested in sociolinguistic analysis in future studies.

**Funding:** This research received no external funding.

**Institutional Review Board Statement:** The study was conducted in accordance with the Declaration of Helsinki, and approved by the Human Research Ethics Committee of The University of Hong Kong (protocol code EA180614) on 19 June 2018.

**Informed Consent Statement:** Informed consent was obtained from all subjects involved in the study.

**Data Availability Statement:** The data presented in this study are available in Appendix A and the author's PhD thesis (Liao 2023).

**Conflicts of Interest:** The authors declare no conflict of interest.

## Abbreviations

| In the text | | In tables and figures | |
|---|---|---|---|
| EMC | Early Middle Chinese | CT | Central Tai |
| LMC | Late Middle Chinese | KD | Kra-Dai (aka Tai-Kadai) |
| MC | Middle Chinese | KS | Kam-Sui |
| MSEA | Mainland Southeast Asia | KT | Kam-Tai |
| OC | Old Chinese | NT | Northern Tai |
| | | NZ | Northern Zhuang |
| | | SWT | Southwestern Tai |
| | | YNZ | Yongnan Zhuang |
| | | YZ | Yang Zhuang |

## Appendix A. Survey of the Development of the Chinese Characters with *shǎng* Tone Conditioned by an EMC Voiced Obstruent Initial in Some Modern Lingnan Languages and Putonghua (中古漢語全濁上聲字發展情況調查表)

| Tone Sinitic/Tai (Language) | | 平話 Pinghua | | | 德保侬壯 Debao YZ | 香港粵語 Cantonese (HK) | | 官話 Mandarin | |
|---|---|---|---|---|---|---|---|---|---|
| | | 橫縣 Hengxian | 亭子 Tingzi | 位子祿 Weizilu | 老借音 MC loanword | 白讀 Colloquial | 文讀 Literary | 柳州 Guiliu | 普通話 Putonghua |
| 陰平 A1 | A1U/A1ACU | 34 | 41 | 53 | 453 | 55 | 55 | 44 | 55 |
| | A1A/A1G | | | | 31 | | | | |
| 陽平A2 | | 213 | 21 | 21 | 31 | 21 | 21 | 31 | 35 |
| 陰上B1/C1 | | 33 | 33 | 33 | 24? | 25 | 25 | 54 | 214 |
| 次濁上B2S/C2 | | 22 | 13 | 13 | 213? | 13 | 13 | 54 | 214 |
| 全濁上B2O/C2 | | 22 | 13 | 13 | | 13 | 22 | 25 | 51 |
| 陰去 C1 | C1U/B1UC | 55 | 55 | 55 | 55 | 33 | 33 | 25 | 51 |
| | C1A/B1AG | | | 35 | 33 | | | | |
| 濁去C2 | | 42 | 223 | 22 | 33 | 22 | 22 | 245 | 51 |
| 長入 DL1 (陰) | DL1UC | 33 | 33 | 33 | 55 | 33 | 33 | 31 入派陽平 *rù* shifted to the lower *píng* tone | 55, 35, 214, 51入派四聲 *rù* shifted to one of the other four tones |
| | DL1AG | | | | 33 | | | | |
| | 次濁DL2S | 42 | 13 | 23 | 33 | 22 | 22 | | |
| | 全濁DL2O | 42 | 22 | 22 | | 22 | 22 | | |
| 短入 | 陰DS1 | 55 | 33 | 33 | 45 | 55 | 55 | | |
| | 次濁DS2S | 11 | 13 | 23 | 21 | 22 | 22 | | |
| | 全濁DL2S | 11 | 22 | 22 | | 22 | 22 | | |

| | | | | | | | | | |
|---|---|---|---|---|---|---|---|---|---|
| 並母 | 部 | pəw$^{22}$ | pu$^{223}$ | pəu$^{22}$ | pəw$^{33}$ | - | pɔw$^{22}$ | pu$^{25}$ | pu$^{51}$ |
| | 簿 | pəw$^{22}$ | pu$^{223}$ | pəu$^{22}$ | pəw$^{33}$ | - | pɔw$^{22}$ | pu$^{25}$ | pu$^{51}$ |
| | 罷 | pa$^{42}$ | pa$^{223}$ | pa$^{22}$ | pa:$^{33}$ | - | pa$^{22}$ | pa$^{25}$ | pa$^{51}$ |
| | 倍 | pu$^{22}$ | puj$^{13}$ | pɔi$^{13}$ | po:j$^{33}$ | pʰuj$^{13}$ | - | pəj$^{25}$ | pei$^{51}$ |
| | 被 | pəj$^{22}$ | pi$^{13}$ | pəi$^{13}$ | pəj$^{33}$ | pʰɛj$^{13}$ | pɛj$^{22}$ | pəj$^{25}$ | pei$^{51}$ |
| | 婢 | pəj$^{42}$ | pi$^{223}$ | pəi$^{22}$ | # pɔ:j$^{213?}$ | pʰɛj$^{13}$ | - | pi$^{25}$ | pi$^{51}$ |
| | 陛 | pəj$^{42}$ | pi$^{223}$ | pəi$^{55}$ | paj$^{33}$ | - | paj$^{22}$ | pi$^{25}$ | pi$^{51}$ |
| | 抱 | pɛw$^{22}$ | paw$^{13}$ | pau$^{13}$ | pa:w$^{33}$ | pʰow$^{13}$ | - | pɑ$^{25}$ | pau$^{51}$ |
| | 鮑 | pøw$^{55}$ | paw$^{41}$ | pau$^{53}$ | - | paw$^{55}$ | paw$^{22}$ | pɑ$^{25}$ | pau$^{51}$ |
| | 鰾 | piw$^{34}$ | - | - | - | pʰiw$^{13}$ | - | piɑ$^{25}$ | piau$^{51}$ |
| | 辯 | pin$^{42}$ | pin$^{223}$ | pin$^{22}$ | pi:n$^{33}$ | - | pin$^{22}$ | pjẽ$^{25}$ | piɛn$^{51}$ |
| | 辨 | pin$^{42}$ | pin$^{223}$ | pin$^{22}$ | pi:n$^{33}$ | - | pin$^{22}$ | pjẽ$^{25}$ | piɛn$^{51}$ |
| | 辮 | pin$^{22}$ | pɛn$^{13}$ | pɛn$^{13}$ | pi:n$^{33}$ | pin$^{55}$ | - | pjẽ$^{25}$ | piɛn$^{51}$ |
| | 伴 | pun$^{22}$ | pun$^{13}$ | pun$^{22}$ | pu:n$^{33}$ | pʰun$^{13}$ | pun$^{22}$ | pã$^{25}$ | pan$^{51}$ |
| | 拌 | pun$^{22}$ | pun$^{13}$ | pun$^{22}$ | pu:n$^{33}$ | - | pun$^{22}$ | pã$^{25}$ | pan$^{51}$ |
| | 笨 | pən$^{42}$ | pɐn$^{223}$ | pɐn$^{22}$ | pan$^{33}$ | - | pɐn$^{22}$ | pã$^{25}$ | pən$^{51}$ |
| | 棒 | paŋ$^{22}$ | paŋ$^{223}$ | paŋ$^{22}$ | pu:ŋ$^{33}$ | pʰaŋ$^{13}$ | - | pɑŋ$^{25}$ | paŋ$^{51}$ |
| | 蚌 | pøŋ$^{42}$ | paŋ$^{13}$ | paŋ$^{13}$ | pu:ŋ$^{33}$ | pʰɔŋ$^{13}$ | - | pɑŋ$^{25}$ | paŋ$^{51}$ |
| | 並 | pɐŋ$^{42}$ | pɐŋ$^{55}$ | pən$^{55}$ | # pjam$^{33}$ | - | piŋ$^{22}$ | pɪŋ$^{25}$ | piŋ$^{51}$ |
| 奉母 | 父 | fəw$^{22}$ | fu$^{21}$ | fu$^{22}$ | # po:$^{33}$ | - | fu$^{22}$ | fu$^{25}$ | fu$^{51}$ |
| | 腐 | fəw$^{22}$ | fu$^{21}$ | fu$^{22}$ | fəw$^{33}$ | - | fu$^{22}$ | fu$^{54}$ | fu$^{214}$ |
| | 輔 | pʰəw$^{33}$ | pu$^{33}$ | phu$^{33}$ | fəw$^{33}$ | - | fu$^{22}$ | fu$^{54}$ | fu$^{214}$ |
| | 婦 | fəw$^{22}$ | fu$^{223}$ | fu$^{22}$ | - | fu$^{13}$ | - | fu$^{25}$ | fu$^{51}$ |
| | 負 | fəw$^{42}$ | fu$^{223}$ | fu$^{22}$ | - | - | fu$^{22}$ | fu$^{25}$ | fu$^{51}$ |
| | 阜 | pəw$^{42}$ | pu$^{223}$ | fəu$^{22}$ | # pəw$^{33}$ | - | fɐw$^{22}$ | fu$^{25}$ | fu$^{51}$ |
| | 范 | fam$^{55}$ | fan$^{55}$ | fam$^{55}$ | fa:m$^{33}$ | - | fan$^{22}$ | fɐn$^{25}$ | fan$^{51}$ |
| | 範 | fam$^{55}$ | fan$^{55}$ | fam$^{55}$ | fa:m$^{33}$ | - | fan$^{22}$ | fã$^{25}$ | fan$^{51}$ |
| | 犯人<br>犯法 | fam$^{22}$ | fam$^{13}$<br>fam$^{223}$ | fam$^{22}$ | fa:m$^{33}$ | - | fan$^{22}$ | fã$^{25}$ | fan$^{51}$ |
| | 憤 | fən$^{55}$ | fɐn$^{13}$ | fɐn$^{13}$ | fan$^{33}$ | fɐn$^{13}$ | - | fən$^{25}$ | fən$^{51}$ |
| | 忿 | fən$^{55}$ | fɐn$^{13}$ | fɐn$^{13}$ | fan$^{33}$ | fɐn$^{13}$ | fɐn$^{22}$ | fən$^{25}$ | fən$^{51}$ |
| | 奉 | foŋ$^{42}$ | føŋ$^{223}$ | fʊŋ$^{22}$ | fuŋ$^{33}$ | - | fuŋ$^{22}$ | fuŋ$^{25}$ | fəŋ$^{51}$ |
| 定母 | 舵 | tø$^{213}$ | ta$^{13}$ | ta$^{35}$ | tɔ:$^{33}$ | tʰaj$^{13}$ | tʰɔ$^{21}$ | to$^{25}$ | tuo$^{51}$ |
| | 惰 | tøi$^{42}$ | tɔ$^{223}$ | tɔ:$^{22}$ | tɔ:$^{33}$ | - | tɔ$^{22}$ | to$^{25}$ | tuo$^{51}$ |
| | 肚 | təw$^{33}$ | tu$^{33}$ | tɔ:$^{13}$ | to$^{33}$ | tʰow$^{13}$ | - | tu$^{25}$ | tu$^{51}$ |
| | 杜 | təw$^{42}$ | tu$^{223}$ | tɔ:$^{22}$ | to$^{33}$ | - | tɔw$^{22}$ | tu$^{25}$ | tu$^{51}$ |
| | 代 | taj$^{22}$ | taj$^{223}$ | tai$^{22}$ | ta:i$^{31}$ | - | tɔj$^{22}$ | taj$^{25}$ | tai$^{51}$ |
| | 待 | taj$^{22}$ | taj$^{223}$ | tai$^{22}$ | ta:j$^{33}$ | - | tɔj$^{22}$ | taj$^{25}$ | tai$^{51}$ |
| | 怠 | taj$^{22}$ | taj$^{223}$ | - | ta:j$^{33}$ | tʰɔj$^{13}$ | - | taj$^{25}$ | tai$^{51}$ |
| | 殆 | taj$^{22}$ | taj$^{223}$ | tai$^{22}$ | ta:j$^{33}$ | tʰɔj$^{13}$ | - | taj$^{25}$ | tai$^{51}$ |
| | 弟 | tej$^{22}$ | tɛj$^{223}$ | tɐi$^{13}$ | taj$^{33}$ | - | tɐj$^{22}$ | ti$^{25}$ | ti$^{51}$ |
| | 道 | tɛw$^{42}$ | taw$^{223}$ | tau$^{22}$ | ta:w$^{33}$ | - | tɔw$^{22}$ | tɑ$^{25}$ | tau$^{51}$ |
| | 稻 | tɛw$^{42}$ | taw$^{223}$ | tau$^{22}$ | ta:w$^{33}$ | - | tɔw$^{22}$ | tɑ$^{25}$ | tau$^{51}$ |
| | 淡 | tam$^{22}$ | tam$^{13}$ | tam$^{13}$ | ta:m$^{33}$ | tʰam$^{13}$ | tam$^{22}$ | tã$^{25}$ | tan$^{51}$ |
| | 簞 | - | - | tan$^{53}$ | # te:m$^{33}$ | tʰim$^{13}$ | - | tẽ$^{25}$ | tiɛn$^{51}$ |
| | 誕 | tan$^{55}$ | tan$^{55}$ | tan$^{55}$ | ta:n$^{33}$ | - | tan$^{33}$ | tã$^{25}$ | tan$^{51}$ |
| | 艇 | tʰəŋ$^{33}$ | taŋ$^{33}$ | tʰɛŋ$^{13}$ | tʰəŋ$^{24?}$ | tʰɛŋ$^{13}$ | - | tʰin$^{54}$ | tʰin$^{214}$ |
| | 挺 | tʰəŋ$^{33}$ | tʰɐŋ$^{33}$ | tʰən$^{33}$ | tʰəŋ$^{24?}$ | tʰin$^{13}$ | - | tʰin$^{54}$ | tʰin$^{214}$ |

| 母 | 字 | | | | | | | | |
|---|---|---|---|---|---|---|---|---|---|
| | 錠 | tən²² | tɛŋ²²³ | tən²² | # te:ŋ³³ | - | tiŋ³³ | tin²⁵ | tiŋ⁵¹ |
| | 斷 | tun²² | tun¹³ | tun¹³ | tu:n³³ | tʰyn¹³ | tyn²² | tʷã²⁵ | tuan⁵¹ |
| | 盾 | tən⁴² | tɛŋ²²³ | tɛn²² | tan³³ | tʰun¹³ | - | tən²⁵ | tuən⁵¹ |
| | 囤 | tən³⁴ | tɛn²¹ | tɛn²¹ | tan³³ | - | tœn²² | tən²⁵ | tuən⁵¹ |
| | 蕩 | tøŋ²² | taŋ²²³ | taŋ²² | ta:ŋ³³ | - | tɔŋ²² | taŋ²⁵ | taŋ⁵¹ |
| | 動 | toŋ²² | tøŋ²²³ | tʊŋ²² | tuŋ³³ | - | tuŋ²² | tuŋ²⁵ | tʊŋ⁵¹ |
| 澄母 | 苧 | - | tɕy³³ | tʃəi¹³ | tɕøy³³ | tsʰy¹³ | - | tsu²⁵ | tʂu⁵¹ |
| | 柱 | tsu²² | tɕy¹³ | tʃəi¹³ | tɕøy³³ | tsʰy¹³ | - | tsu²⁵ | tʂu⁵¹ |
| | 雉 | - | tɕi⁴¹ | tʃi²² | tɕəj³³ | - | tsi²² | tsɿ²⁵ | tʂʅ⁵¹ |
| | 痔 | tsi⁴² | tɕi²²³ | tʃi²² | tɕəj³³ | - | tsi²² | tsɿ²⁵ | tʂʅ⁵¹ |
| | 峙 | tsi²² | - | - | tɕəj³³ | tsʰi¹³ | tsi²² | tsɿ²⁵ | tʂʅ⁵¹ |
| | 趙 | tsiw²² | tɕiw²²³ | tʃiu²² | tɕiw³³ | - | tsiw²² | tsɑ²⁵ | tʂau⁵¹ |
| | 兆 | tsiw⁴² | tɕiw²²³ | tʃiu²² | tɕiw³³ | - | siw²² | tsɑ²⁵ | tʂau⁵¹ |
| | 肇 | siw⁴² | - | - | tɕiw³³ | - | siw²² | tsɑ²⁵ | tʂau⁵¹ |
| | 紂 | - | tɕɐw²²³ | tʃɐu²² | tɕaw³³ | - | tsɐw²² | tsɑ²⁵ | tʂou⁵¹ |
| | 朕 | - | - | - | tɕam³³ | - | tsɐm²² | tsəw²⁵ | tʂən⁵¹ |
| | 篆 | tsun⁵⁵ | ɬyn²²³ | tʃʷan⁵⁵ | tɕy:n³³ | - | ʃyn²² | tsʷã²⁵ | tʂuan⁵¹ |
| | 丈 | tsɛŋ²² | tɕɛŋ²²³ | tʃɛŋ²² | tɕy:ŋ³³ | - | tsœŋ²² | tsɑŋ²⁵ | tʂaŋ⁵¹ |
| | 仗 | tsɛŋ⁵⁵ | tɕɛŋ⁵⁵ | tʃɛŋ²² | tɕy:ŋ³³ | - | tsœŋ²² | tsɑŋ²⁵ | tʂaŋ⁵¹ |
| | 杖 | tsɛŋ⁵⁵ | tɕɛŋ²²³ | tʃɛŋ⁵⁵ | # taw²¹³ʔ | - | tsœŋ²² | tsɑŋ²⁵ | tʂaŋ⁵¹ |
| | 仲 | tsoŋ⁴² | tɕøŋ²²³ | tʃʊŋ²² | tɕʊ:ŋ³³ | - | tsuŋ²² | tsuŋ²⁵ | tʂʊŋ⁵¹ |
| | 輕重 | tsoŋ²² | tɕøŋ¹³ | tʃʊŋ¹³ | tɕuŋ³³ | tsʰuŋ¹³ | tsuŋ²² | tsuŋ²⁵ | tʂʊŋ⁵¹ |
| 從母 | 坐 | tsəw²² | tɕu¹³ | tʃu¹³ | tɕɔ:³³ | tsʰɔ¹³ | tsɔ²² | tso²⁵ | tsuo⁵¹ |
| | 聚 | tsu²² | tɕy²²³ | tʃi²² | tɕøɥ³³ | - | tsœy²² | tsy²⁵ | tɕy⁵¹ |
| | 在 | tsaj²² | tɕaj²²³ | tʃai²²/tʃɐi²² | tɕa:j³³ | - | tsɔj²² | tsaj²⁵ | tsai⁵¹ |
| | 載 | tsaj³³ | tɕaj⁵⁵ | tʃai⁵⁵ | tɕa:j³³ | tsɔj²⁵ | tsɔj³³ | tsaj²⁵ | tsai⁵¹ |
| | 薺 | tsɐj²² | tɕɛj²¹ | tʃɐi²¹ | - | tsʰɐi²⁴ | - | tsi²⁵ | tɕi⁵¹ |
| | 罪 | tsu⁵⁵ | tɕuj²²³ | tʃɔi²² | tɕɔ:j³³ | - | tsœy²² | tsʷəj²⁵ | tsai⁵¹ |
| | 皂 | tsew⁴² | tɕaw²²³ | tʃau²² | tɕa:w³³ | - | tsɔw²² | tsɑ²⁵ | tsau⁵¹ |
| | 造 | tsɛw²² | tɕaw²²³ | tʃau²² | tɕa:w³³ | tsʰow¹³ | tsɔw²² | tsɑ²⁵ | tsau⁵¹ |
| | 漸 | tsim⁴² | tɕim²²³ | tʃim¹³ | tɕi:n³³ | - | tsim²² | tsẽ²⁵ | tɕiɛn⁵¹ |
| | 踐 | tsin⁴² | tɕin³³ | tʃin³³ | tɕi:n³³ | tsʰin¹³ | tsin²² | tsẽ²⁵ | tɕiɛn⁵¹ |
| | 盡 | tsən²² | tɕiɐn²²³ | tʃɐn²² | tɕan³³ | - | tsœn²² | tsin²⁵ | tɕin⁵¹ |
| | 靖 | tɕɛŋ²² | tɕɛŋ²²³ | tʃən²² | tɕəŋ³³ | - | tsiŋ²² | tsin²⁵ | tɕiŋ⁵¹ |
| | 靜 | tɕɛŋ²² | tɕɛŋ²²³ | tʃən²² | tɕəŋ³³ | - | tsiŋ²² | tsin²⁵ | tɕiŋ⁵¹ |
| 邪母 | 序 | tsu²² | tɕy¹³ | tʃi²² | tɕøy³³ | - | tsœy²² | sʲy²⁵ | ɕy⁵¹ |
| | 敘 | tsu²² | tɕy¹³ | tʃi²² | tɕøy³³ | - | tsœy²² | sʲy²⁵ | ɕy⁵¹ |
| | 緒 | tsu²² | ɬy⁵⁵ | ɬi⁵⁵ | tɕøy³³ | sœy¹³ | - | sʲy²⁵ | ɕy⁵¹ |
| | 祀 | si²² | ki²²³ | tʃi²² | tɕəj³³ | - | tsi²² | sɿ²⁵ | sɿ⁵¹ |
| | 巳 | - | ɕi²²³ | tʃi²² | tɕəj³³ | - | tsi²² | sɿ²⁵ | sɿ⁵¹ |
| | 似 | tsɯj⁴² | tɕi³³ | tʃʰi³⁵ | tɕəj³³ | tsʰi¹³ | - | sɿ²⁵ | sɿ⁵¹ |
| | 象 | tsɛŋ²² | tɕɛŋ²²³ | tʃɛŋ²² | # tɕa:ŋ²¹³ʔ | - | tsœŋ²² | sʲaŋ²⁵ | ɕiaŋ⁵¹ |
| | 像 | tsɛŋ²² | tɕɛŋ²²³ | tʃɛŋ²² | tɕy:ŋ³³ | - | tsœŋ²² | sʲaŋ²⁵ | ɕiaŋ⁵¹ |
| | 橡 | tsɛŋ²² | tɕɛŋ²²³ | tʃɛŋ²² | tɕy:ŋ³³ | - | tsœŋ²² | sʲaŋ²⁵ | ɕiaŋ⁵¹ |
| 崇母 | 士 | si⁴² | ɕɛj²²³ | ɬɐi²² | # ɬaj⁵⁵ | - | si²² | sɿ²⁵ | ʂʅ⁵¹ |
| | 仕 | si⁴² | ɕɛj²²³ | ɬɐi²² | # ɬaj⁵⁵ | - | si²² | sɿ²⁵ | ʂʅ⁵¹ |
| | 柿 | səj²² | ɕɛj²²³ | ɬɐi²² | ɬaj³³ | tsʰi¹³ | - | sɿ²⁵ | ʂʅ⁵¹ |
| | 撰 | - | lyn³³ | tʃan⁵⁵ | tɕɔ:n³³ | - | tsan²² | tsʷã²⁵ | tʂuan⁵¹ |

| 聲母 | 字 | | | | | | | | |
|---|---|---|---|---|---|---|---|---|---|
| 船母 | 甚 | - | - | - | - | - | sɐm²² | sən²⁵ | ʂən⁵¹ |
| | 舓 | si⁴² | tim²¹? | kɛm³⁵? | - (li:³¹?) | saj¹³ | - | sɿ²⁵ | ʂʅ⁵¹ |
| 禪母 | 社 | si²² | ɕɛ¹³ | ɬɛ¹³ | # tɕy:²¹³? | sɛ¹³ | - | siͤ²⁵ | ʂə⁵¹ |
| | 墅 | suj⁵⁵ | - | - | ɬøy³³ | sœy¹³ | sœy²² | su²⁵ | ʂu⁵¹ |
| | 豎 | su⁴² | ɕy³³ | ɬəi²² | ɬøy³³ | - | sy²² | su²⁵ | ʂu⁵¹ |
| | 是 | si²² | ɕi²²³ | ɬi²² | # tɕøy³³ | - | si²² | sɿ²⁵ | ʂʅ⁵¹ |
| | 氏 | si⁴² | ɕi²²³ | ɬi²² | # te:³³ | - | si²² | sɿ²⁵ | ʂʅ⁵¹ |
| | 市 | si²² | ɕi¹³ | ɬi¹³ | ɬøy³³ | si¹³ | - | sɿ²⁵ | ʂʅ⁵¹ |
| | 恃 | si²² | ɕi¹³ | tʃʰi³⁵ | ɬøy³³ | tsʰi¹³ | - | sɿ²⁵ | ʂʅ⁵¹ |
| | 紹 | siw⁴² | ɕiw²²³ | ɬiu²² | ɬiw³³ | - | siw²² | sɑ²⁵ | ʂau⁵¹ |
| | 受 | səw²² | ɕɐw²²³ | ɬəu²² | ɬaw³³ | - | sɐw²² | səw²⁵ | ʂou⁵¹ |
| | 甚 | səm⁴² | ɕɐm³³ | ɬɐm²² | ɬan³³ | - | sɐm²² | sən²⁵ | ʂən⁵¹ |
| | 善 | sin⁴² | - | ɬin²² | ɬi:n³³ | - | sin²² | siᵉ²⁵ | ʂan⁵¹ |
| | 腎 | sən²² | ɕʲɐn¹³ | ɬɐn¹³ | ɬan³³ | sɛn¹³ | sɐn²² | sən²⁵ | ʂən⁵¹ |
| | 上 | sən²² | ɕən¹³ | ɬɐŋ¹³ | ɬy:ŋ³³ | sœn¹³ | sœn²² | sɑŋ²⁵ | ʂaŋ⁵¹ |
| 群母 | 巨 | ku⁴² | ky¹³ | kəi⁵⁵ | køɥ³³ | - | kœy²² | ky²⁵ | tɕy⁵¹ |
| | 拒 | - | ky¹³ | kəi⁵⁵ | køɥ³³ | kʰœy¹³ | - | ky²⁵ | tɕy⁵¹ |
| | 距 | ku⁴² | ky¹³ | kəi⁵⁵ | køɥ³³ | kʰœy¹³ | - | ky²⁵ | tɕy⁵¹ |
| | 技 | kɯj⁴² | ki²²³ | kəi²² | kəj³³ | - | kɛj²² | ki²⁵ | tɕi⁵¹ |
| | 妓 | kɯj⁴² | ki²²³ | kəi³³ | kəj³³ | - | kɛj²² | ki²⁵ | tɕi⁵¹ |
| | 徛 | kɯj³³ | - | ki²¹ | - | kʰɛj¹³ | - | ki²⁵ | tɕi⁵¹ |
| | 跪 | kʷəj⁴² | kʷɛj²²³ | kʷəi²² | kʷəj³³ | - | kʷɐj²² | kʷəj²⁵ | kuei⁵¹ |
| | 舅 | tsu²² | kɐw¹³ | kəu¹³ | kʲaw³³ | kʰɐw¹³ | - | kʲəw²⁵ | tɕiou⁵¹ |
| | 臼 | - | kɐw¹³ | kɐu⁵⁵ | # kʲɔ:k²¹ | kʰɐw¹³ | - | kʲəw²⁵ | tɕiou⁵¹ |
| | 咎 | tsiw³⁴ | kɐw⁵⁵ | kɐu⁵⁵ | - | - | kɐw³³ | kʲəw²⁵ | tɕiou⁵¹ |
| | 儉 | kim⁴² | kim²²³ | kim²² | kim²⁴? | - | kim²² | kẽ⁵⁴ | tɕiɛn²¹⁴ |
| | 件 | kin²² | kyn²²³ | kin²² | ki:n³³ | - | kin²² | kẽ²⁵ | tɕiɛn⁵¹ |
| | 鍵 | kin⁴² | kin⁵⁵ | kim¹³ | ki:n³³ | - | kin²² | kẽ²⁵ | tɕiɛn⁵¹ |
| | 勉強 | kʰɛŋ³³ | kʰɛŋ³³ | kʰɛŋ³³ | ky:ŋ³³ | kʰœŋ¹³ | - | kʰʲaŋ⁵⁴ | tɕʰian²¹⁴ |
| | 倔強 | kʰɛŋ³³ | kʰɛŋ³³ | kɛŋ²¹ | ky:ŋ³³ kʲo:ŋ³¹ | kʰœŋ¹³ | kœŋ²² | kʲaŋ²⁵ | tɕian⁵¹ |
| | 豬圈 | kʰun³⁴ | hyn⁴¹ | kʷʰin⁵³ | ku:n³³ | - | ky:n²² | kʷẽ²⁵ | tɕyɛn⁵¹ |
| | 近 | kən²² | kɐn¹³ | kɛn¹³ | kan³³ | kʰɐn¹³ | kɐn²² | kʲin²⁵ | tɕin⁵¹ |
| | 窘 | - | - | kʷən²² | kun³³ | - | kʰʷɐn³³ | kʲuŋ⁵⁴ | tɕiʊŋ²¹⁴ |
| | 菌 | kun²² | kʰʷɐn¹³ | kʷən³⁵ | kun³³ | - | kʰʷɐn³⁵ | kyn²⁵ | tɕyn⁵¹ |
| 匣母 | 禍 | həw⁴² | wu²²³ | hu²² | wa:³³ | - | wɔ²² | ho²⁵ | xuo⁵¹ |
| | 下 | ja²² | ja²²³ | ja²² ha¹³ | ja:⁵⁵ | ha¹³ | ha²² | hʲa²⁵ | ɕia⁵¹ |
| | 戶 | həw²² | hu²²³ | hɔ²² | wu:³³ | - | wu²² | hu²⁵ | xu⁵¹ |
| | 滬 | həw⁴² | hu²²³ | hɔ²² | wu:³³ | - | wu²² | hu²⁵ | xu⁵¹ |
| | 亥 | hø²² | haj²²³ | hai²² | ja:j³³ | - | hɔj²² | haj²⁵ | xai⁵¹ |
| | 駭 | - | - | hai²² | ja:j³³ | haj¹³ | - | haj²⁵ | xai⁵¹ |
| | 蟹 | - | haj¹³ | hai³³ | - | haj¹³ | - | haj⁵⁴ | ɕiɛ⁵¹ |
| | 解姓 | tsaj³³ | kaj³³ | - | ja:j³³ | - | haj²² | haj²⁵ | ɕiɛ⁵¹ |
| | 匯 | wei⁴² | wɛj²²³ | wəi²² | wo:j³³ | - | wuj²² | hʷəj²⁵ | xuei⁵¹ |
| | 浩 | hɛw³³ | haw⁵⁵ | hau³³ | ja:w³³ | - | hɔw²² | hɑ²⁵ | xau⁵¹ |
| | 後 | hɔw⁴² | hɐw²²³ | hɐu²² | jaw³³ | - | hɐw²² | həw²⁵ | xou⁵¹ |
| | 后 | hɔw⁴² | hɐw²²³ | hɐu²² | jaw³³ | - | hɐw²² | həw²⁵ | xou⁵¹ |
| | 厚 | hɔw²² | hɐw²²³ | hɐu¹³ | jaw³³ | hɐw¹³ | - | həw²⁵ | xou⁵¹ |
| | 憾 | høm³³ | hɐm²²³ | ham²² | ja:m³³ | - | hɐm²² | hã²⁵ | xan⁵¹ |

| | | | | | | | | |
|---|---|---|---|---|---|---|---|---|
| 艦 | lam42 | lam13 | lam22 | - | - | lam22 | kẽ25 | tɕiɛn51 |
| 旱 | høn22 | han13 | han13 | ja:n33 | hɔn13 | - | hã25 | xan51 |
| 限 | han22 | han223 | han22 | ja:n33 | - | han22 | hã25 | ɕiɛn51 |
| 很 | hɐn213 | hɐn33 | hɐn33 | han24ʔ | - | hɐn25 | hən54 | xən214 |
| 緩 | wun22 | wan33 | wan33 | wu:n55 | - | wun22 | hʷã54 | xuan214 |
| 皖 | wən33 | wan33 | wan33 | wa:n33 | wun13 | - | wã25 | wan51 |
| 晃 | ʔuŋ55 | - | kʷʰaŋ33 | wu:ŋ33 | - | fɔŋ25 | hʷaŋ25 | xuan51 |
| 混 | ŋʷən213 | kʰʷən33 | kʰʷʰən33 | kʰʷan24ʔ | - | wɐn22 | hʷən25 | xuən51 |
| 項 | haŋ42 | haŋ223 | haŋ22 | ja:ŋ33 | - | hɔŋ22 | hʲaŋ25 | ɕiaŋ51 |
| 杏 | haŋ42 | hɐŋ55 | hən35 | jaŋ33 | - | hɐŋ22 | hʲin25 | ɕiŋ51 |
| 幸 | haŋ42 | haŋ223 | hɛŋ22 | jeŋ33 | - | hɐŋ22 | hʲin25 | ɕiŋ51 |
| 汞 | koŋ55 | køŋ33 | kʊŋ55 | kuŋ24ʔ | - | huŋ22 | kuŋ54 | kʊŋ214 |
| 迥 | - | - | - | kʷəŋ24ʔ | - | kʷin25 | kʲuŋ54 | tɕiʊŋ214 |

[1] Note. The main purpose of this wordlist is to identify the percentage of Chinese characters' *zhuóshǎng guī qù* (濁上歸去: the *shǎng* tone conditioned by original voiced obstruent initials shifted to the lower *qù* tone) phenomena in the phonological history of the Chinese language, among the different Lingnan Sinitic languages/dialects. This appendix contains a total of 156 characters with original voiced obstruent initials and tone *shǎng*. (The character '強' is divided into two items based on the different meanings of 'reluctant' and 'stubborn'). The forms in the appendix are based on my own transcription based on phonological analysis (e.g., the rimes /ik/, /in/, and /iŋ/ in Cantonese are actually phonetically [ɪk], [ɪn], and [ɪŋ], respectively), except for Nanning Weizilu Pinghua, which was adopted from de Sousa (Forthcoming). The sources of the data in this appendix are as follows. Nanning Weizilu Pinghua was adopted from de Sousa (Forthcoming); Binyang Luxu Pinghua and Nanning Tingzi Pinghua were basically adapted from Li (2000) and transcribed in my system, and many of the missing items in Binyang Luxu Pinghuar were supplemented by me based on my own fieldwork; data of Hong Kong Cantonese, Liuzhou Urban variety of Guiliu Mandarin, and Standard Mandarin were collected by myself.

## Notes

[1] The core Lingnan region in this paper refers to those areas centered on western Guangdong Province to the west of the Pearl River Delta (including Guangzhou, Hong Kong, and Macau) and the entirety of Guangxi. Some of the exclusive linguistic areal traits, such as the sole CVX$^T$ syllable structure and its correlative phonological features, are only found in this core Lingnan area (Liao 2023), which excludes Southern Hunan and Southwestern Jiangxi Provinces, as well as the Southern Min-speaking areas of the eastern part of Guangdong Province, Hainan Island outside the mainland, and northern Vietnam to the south of the Sino–Vietnamese border, which were also historically regarded as part of Lingnan.

[2] At least for the proto-tone system, there is no tonal contrast on checked syllables, so the so-called tonal category *D, which only "exists" on checked syllables, is not, strictly speaking, a tone. However, in traditional Chinese phonology, a checked syllable is also considered to be equivalent to a tonal category, "entering tone" (tone *rù*), as each syllable in tonal languages is generally considered to be loaded with a tone. Moreover, in the later register tonal split, *D was also split into modern tones with different pitch levels. For convenience, we follow the standard practice of treating proto-tonal category *D as a tone.

[3] In the mainstream circles studying the tonal system of the languages of China, it is common to use odd and even numbers to label the upper- and lower-register tones, respectively. Therefore, A1, B1, C1, and D1 denote upper-register tones conditioned by the original voiceless initials, while A2, B2, C2, and D2 denote lower-register tones conditioned by the original voiced initials. If cardinal numerals are used to mark tones, 1, 3, 5, and 7 correspond to A1, B1, C1, and D1, respectively, and 2, 4, 6, and 8 correspond to A2, B2, C2, and D2, respectively. In addition, in modern tonal languages, it is possible for the original upper and lower registers to be switched between high and low pitches, known as "tonal flip-flops" (Matisoff 1973; Fu 1995, p. 82). Tonal flip-flop Tai languages, such as most Southwestern Tai varieties, including Standard Thai, are mostly found in the "new territories of Tai emigrations", and this is one of the reasons for considering tonal flip-flop as a secondary development (Liao 2016b, p. 100).

[4] Glottalized voiced stops /ʔb-/ and /ʔd-/ are sometimes described as implosive stops /ɓ-/ and /ɗ-/, respectively, but because it is the preglottalized segment ʔ- that conditioned primary tonal split in history, it is necessary to consistently assign them as preglottalized sounds, phonologically speaking (cf. Liao 2022, p. 4).

[5] Voicing alternation of Tai languages refers to the problem that, in a group of cognate Tai etyma, the Southwestern/Central Tai dialects consistently have upper-register tones reflecting proto-original voiceless initial consonants, but the Northern Tai dialects have lower-register tones reflecting proto-voiced initial consonants (cf. Li 1966, 1970, 1977, pp. 36–39; Gedney 1989; Diller 1998, p. 7; Thurgood 2002, 2007; Pittayaporn 2009, p. 13; Liao 2016a, p. 80; 2016b, pp. 16–20, 27–28, 76–77, 141–50, 187–89; 2022,

pp. 8–10). For example, the Thai term /hu:$^{24}$/ 'ear' is located in box A1 in Gedney's formulation, because, in southwestern Tai languages, its tone reflects a proto voiceless initial *kr-, but its cognates in Northern Tai, such as Wuming Zhuang/ɣɯ:$^{31}$/'ear', reflect proto-Tai *r- and should thus be located in box A4 for Northern Tai languages. Thus, it fails to serve as a testing etymon for all Tai languages. In Liao (2016b, 2022), another two series of voicing alternations in Tai languages are identified. One of them is the situation opposite to the above one, i.e., Southwestern/Central Tai dialects consistently have lower-register tones reflecting original voiced initial consonants, but the Northern Tai dialects have upper-register tones reflecting original voiceless initial consonants of proto-Tai (Liao 2016b, pp. 141–42, 187–89; 2022, p. 38). Another one is found in a series of Tai etyma, in which a cluster of Guibei Zhuang varieties (Northern Zhuang), represented by Huanjiang–Suogan, have upper-register tones reflecting the original voiceless initial consonants, but the vast majority of Tai varieties (including the five major Tai subbranches, Northern Zhuang, Yongnan Zhuang, Saek, Central Tai, and Southwestern Tai) have lower-register tones reflecting the original voiced initial consonants of proto-Tai (Liao 2016b, pp. 165–68; 2022, p. 39).

6    These five sets of initial consonants are marked with an asterisk * to indicate that these laryngeal features were present at the time of the tonal split, but the aspirated initial consonants of Central Tai and Southwestern Tai are suggested to be a post-proto-Tai innovation (Liang and Zhang 1996; Pittayaporn 2009; Liao 2016b, pp. 121–24; 2022, p. 14). In addition, after conditioning primary/secondary tonal splits, some initial consonants merged into other consonants or lost their original laryngeal feature, e.g., *$^{h}$m- has lost its pre-aspirated segment and merged into *m-, and *$^{ʔ}$d- has lost its pre-glottalized segment to become /d-/ in modern Thai; however, the tones conditioned by them reveal their laryngeal features by the time of the tonal split. For the full entries of each set, refer to Liao (2022, pp. 25, 37–41).

7    The integrity of this tone-box scheme is further achieved by providing an ancillary chart for how to solve the problem of tonal testing failure due to irregular tonal correspondence, including voicing alternation (Liao 2022, p. 10). This ancillary chart provides nineteen sound series for those etyma with irregular tonal correspondences so that these etyma can be placed in different boxes of the main box according to their respective merging direction in the different Tai subgroups. In this way, there are two types of etyma to place in each tone box. One is "test etyma", which can be directly placed in a fixed box for all Tai subgroups, as shown in Table 7. The other is flexible etyma, which are placed in a box depending on the Tai subgroup to which the language variety under investigation belongs, as different Tai subgroups may have these etyma with different merging directions to the terminal tone box. For example, proto-Tai *ʼk.ɣɤ:$^{A}$ 'ear' (cf. Liao 2022, p. 38) should be put into box A1A for Central Tai and Southwestern Tai, as it first developed to *kru:$^{A}$ in proto-Southern Tai (the direct parent of both Central and Southwestern Tai), and ultimately to have an aspirated initial and an upper-register tone in modern Central/Southwestern Tai subgroups; it should be put into box A2 for all Northern Tai languages (including three subgroups, Northern Zhuang, Yongnan Zhuang, and Saek), because it first developed to *rɯ:$^{A}$ to have a voiced initial in proto-Northern Tai, and ultimately developed to have a lower-register tone in modern Northern Tai dialects. Refer to Liao (2022, pp. 10, 23–27) for more details.

8    Note that, for Guangzhou–Hong Kong Cantonese, only the original tone (whether in the literary and colloquial readings) is used for the items in the wordlist, not the changed tone, which may be more frequently used in spoken language, and the data of Binyang Luxu Pinghua were mostly adopted from Li (2000) and partially from my own field notes.

9    Zhou and Zhu (2020) claimed that, in the Zhajin variety of Gan Chinese, there is a sixteen-tone system and that it is the language with the largest number of tones ever found. However, they did not take into account the complementary distribution of the system, and so considered the tones on the checked syllables to be independent tones. Even if the six tones on the checked syllables they identified are considered to be allotones of the tones on the smooth syllables, the remaining ten tones are also numerous, more than Kra-Dai's largest tone number in Southern Kam, which has nine distinctive tones. In some previous studies on this Zhajin variety of Gan Chinese, there were five to seven tones on smooth syllables according to different authors, but only this experimental acoustic study suggests that there are ten tones on smooth syllables, which I believe is the result of treating some of the slightly natural modulations of each allotone of the same toneme conditioned by different initial categories as separate tones, without taking into account the complementary distribution of the system. However, even if this dialect has 10 tones on the smooth syllable, as they claimed, there are only two secondary registers on the high/upper (清 clear) register: semi-clear (次清 i.e., aspirated surd) and clear (全清 i.e., unaspirated surd); the remaining three secondary registers that have caused possible secondary tonal splits are all of the low/lower (濁 muddy) register: semi-muddy (次濁, i.e., sonorants/liquad), muddy stops (濁塞 i.e., original voiced plosives/affricates), and muddy fricatives (濁擦, i.e., original voiced fricatives). In other words, this Sinitic language, which is claimed to have 16 tones (or 10 distinctive tones), requires two sets of laryngeal features in the upper (clear)-register initials, which is still fewer than the four sets required for the upper (original voiceless) register in Tai. Moreover, although the distribution of Gan Chinese is not in the core areas of the contemporary Lingnan region, its southern varieties are within the northeastern border of the early Lingnan region, not to mention that the whole area south of the Yangtze River is also considered historically an early distribution area for the Kra-Dai family. Therefore, it cannot be ruled out that the secondary tonal split in the upper register of this dialect was also a trait diffused from Kam-Tai in the past.

10    As aforementioned, Debao Urban Yang Zhuang, as a Kra-Dai language, is in the Kra-Dai circle using the letter B to correspond to the *qù* tone and the letter C to correspond to the *shǎng* tone in traditional Chinese phonology, or, in other words, tone *B of Kra-Dai corresponds to *C of Sinitic, and tone *C of Kra-Dai corresponds to *C of Sinitic. However, this part is a discussion of the Sinitic languages and the Sinitic loanword in Yang Zhuang, so for the sake of comparison, the Sinitic tones are also used here in the case of Yang Zhuang, i.e., B refers to *shǎng* and C is *qù*.

[11] However, as Liao (2022, pp. 16, 42) mentioned, there are indeed some Tai etyma involved in tonal categories B (corresponding to Sinitic tone B or *qù*) and C (corresponding to Sinitic tone C or *shăng*) alternation between Northern Tai (Northern Zhuang-Yongnan Zhuang-Saek) and Southern Tai (Central Tai–Southwestern Tai), and it now appears that they may indeed be the result of the effect of the Chinese loanwords in the *zhuóshăng guī qù* series. For example, the tone of the Southern Tai etyma *bi:[B] 'elder siblings' is B2 in modern Central–Southwestern Tai dialects, but in the Northern Tai dialect, it is C2, reflecting *bi:[C]. In another study, Liao attributed this phenomenon to "phonological contamination", i.e., tone C2 of this etymon in Northern Tai is a result of the analogical change, replacing its original tone *B with the tone of another item /nu:ŋ[C2]/ 'younger sibling' in the semantic pair /pi:[C2] nu:ŋ[C2]/ 'siblings (younger and elder brothers)' (Liao 2017, pp. 129–31). However, from the point of view of areal trait, it cannot be ruled out that the original tone of this etyma was *C, as preserved in Northern Tai dialects, but later, under the influence of the Chinese *zhuóshăng guī qù* phenomenon via a massive amount of loanwords, tone *C of this item, which was conditioned by the original voiced stop *b-, changed to tone *B in Southern Tai dialects. However, as tone B/C alternation in Tai is not limited to the lower register (cf. Liao 2022, p. 42), this matter needs to be examined in further research.

[12] It is not too far-fetched either to open the possibility of this being *Wanderwort* (without unknown origin, as it is a relatively common cognitive verb) or from other possible Lingnan substrate languages, such as Hmong-Mien. However, according to the author's observations, generally speaking, if an etymon is borrowed from Hmong-Mien into Sinitic languages, or if it is a Hmong-Mien substratum, it appears to be commonly shared among several Sinitic branches and not limited to Sinitic languages in Lingnan. The word 狗 'dog' (kəu:) in Mandarin and most southern Sinitic languages, for example, is suggested to be borrowed from Hmong-Mien *klu²B and became the primary word for 'dog' in most Sinitic languages after the pre-Han period, with only some Min dialects retaining the Sinitic-origin 犬 'dog' (MC *khiwen:*) in spoken language (Norman 1988, p. 17). In contrast, most of the words of non-Sinitic origin found only in Lingnan Sinitic languages, such as Cantonese, Pinghua, and even Guiliu Mandarin, are more related to Kam-Tai than to Hmomg-Mien. This seems to imply that the close contact with Hmong-Mien during the southward migration of Sinitic languages was mostly completed north of Lingnan, whereas the close contact of Sinitic with non-Sinitic languages in Lingnan was mainly with Kam-Tai instead of Hmong-Mien.

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
