# Peer review of "Tonal Behavior as of Areal and Typological Concerns: Centering on the Sinitic and Kam-Tai Languages in Lingnan"

_languages, doi:10.3390/languages8020148_

Round 1
Reviewer 1 Report
The manuscript deals with an issue of tonal split patterns in Chinese and Tai-Kadai linguistics, which has been repeatedly tested against dialectal data separately in individual Chinese and Tai-Kadai dialects in a ton of previous studies without significantly advancing the methodology and cross-family comparison. This study, however, really tackles the methodological issue of tone paradigm classification, propose an innovative approach and highlight the contribution of Chinese dialects for the historical studies of Tai-Kadai tones and vice versa. It delivers a meritorious contribution to both typological and diachronic studies of languages with complex tone systems, which can serve as a good model for future studies and as an improved guideline for dialectal data collection, particularly advancing the half-century-old Gedney's tone box method to the next level.
Central references are propoerly cited, showing the relevance and coherence of the present study to the ongoing discussion and problems in the research field at a general level. However, the use of terminologies, as this study deals with a large number of languages as well as diferent layers of intermediate proto-languages, is inconsistent in several places. In terms of readability, the text is overall fluent, but some paragraphs are overtly long with a lot of information for readers to process, especially a long list of language names.
As the manuscript contains a lot of specific details on the phenomenon and related theoretical matters, the detailed comments and suggestions are given in the attached pdf-file. At a more general level, several key points are suggested below.
1. In the current version's structure, introduction is overwhelmingly devoted to the presentation of tone boxes. However, that might better be part of Section 2, as it already involves theories related to the classification and division of tone system. Instead of that, the introduction should highlight the methodological aspects of this study more, considering how it improves an inadequate use of tone box theories in previous studies and maximises the method, as well as how the current method applied can reveal new insights and cross-family correlations from the empirical data significantly increasing in the recent decades. Likewise, the conclusions should more explicitly show the methodological merits of this study at a conceptual level, rather than mostly repeating the results from Section 3. These points which are the strength of the current study can be considered for more explicit mentions in both introduction and conclusions.
1.1 The criteria and argumentation for and against inherited features vs. areal traits.
1.2 The hypothesis of lexical diglossia of colloquial vs. literary readings in Lingnan Sinitic, and the approach to deal with this phenomenon as "borrowings" in a conventional sense.
1.3 The tonal evidence for proposing a more fine-grained relative chronology of changes under contact between Sinitic and Kra-Dai, as well as criteria to detect Kam-Tai borrowings in southern Sinitic.
2. Be consistent with where the statement is meant to refer to the whole "Kra-Dai" family, and it just captures the "Kam-Tai" first-level subbranch which is in the focus in this study. There are some statements which either concern the whole family or deal with specific issues not being the case for Kra and Hlai subbranches.
3. There are several claims on some phenomena or typological tendencies, which are not explicitly formulated as the author's own thoughts. In some cases, adding some references might help justifying the validity of the claims.

Author Response
Dear Sir/Madam,
Thank you very much for your comments and suggestions on my paper. I have made some revisions according to the advice from you and the other reviewers. Some of the key points are as follows.
1. I agree with your suggestion that the introduction and conclusion sections of my original manuscript should be more focused on methodological merits. Therefore, I have made substantial changes to these two sections. Moreover, I have moved the parts of the review of tonogenesis and tonal split, which was originally part of the introduction, to Section 2 as you suggested.
2. I have added the Sinitic and Kra-Dai family tree diagrams to the introduction to facilitate understanding of the subgroups of these two language families. I have followed your suggestion to check the whole text to ensure that Kra-Dai and Kam-Tai should be used accurately.
3. I have also mostly adopted your suggestion to add some references to add credibility to some of the claims.
In addition, I have mostly accepted your detailed comments to make changes in my original manuscript.
I have attached a copy of the revised manuscript. The first half shows the "Track Changes" in the revisions, such that be easily viewed by the editors and reviewers. The second half is the full version without track changes, such that you can read the whole manuscript smoothly.
Thank you again for your review!
The author 
13 April 2023

Reviewer 2 Report
This is a very good review of the Gedney tone-box analysis as it applies to Kam-Tai and Sinitic languages within the Lingnan linguistic area. The discussion of secondary tone splits was quite interesting, and the case for secondary splits in the upper register (and between long and short checked syllables) originating in Kam-Tai and diffusing to Yue and Pinghua was compelling. The discussion of secondary splits in the lower register originating in Sinitic languages was also interesting, but a bit less clear; the discussion of Table 18 was the most difficult part for me to follow.
The discussion and arguments in the areal typology discussion are rather lengthy, and it may be possible to simplify this section without obscuring the main points. It is otherwise also very interesting, and I appreciated the discussion about upper limits on the number of tones which requires that tones on checked syllables be understood as allotones (with the useful general rule reiterated that tone categories B = DL and C = DS) as well as the constraint that all four Kam-Tai upper register categories cannot be maintained as distinct, with arguments for an implicational hierarchy of splitting.
Overall, I found this a very useful exposition of tonogenesis as well as areal typology of the Lingnan area and I think this is a valuable contribution to the study of historical linguistics in MSEA.
Author Response
Dear Sir/Madam,
Thank you very much for your comments and suggestions on my manuscript. I have made some revisions to the manuscript according to the advice from you and the other reviewers. Some of the key points are as follows.
1. I have made substantial changes to the introduction and conclusion sections, focusing more on methodological merits. Moreover, I have moved the introduction to tonogenesis and tonal split, which was originally part of Section 1, to Section 2 as this part is relevant to the tone-box theory.
2. I have added the Sinitic and Kra-Dai family tree diagrams to Section 1 to facilitate understanding of the subgroups of these two language families. I have also checked the whole text to ensure the two terms Kra-Dai and Kam-Tai should be accurately used.
I have attached a copy of the revised manuscript. The first half shows the "track changes", such that any changes can be easily viewed by the editors and reviewers. The second half is the full version without “track changes”, so that you can read it smoothly.
Thank you again for your review!
The Author
13 April 2023

Reviewer 3 Report
This paper aims to show secondary tonal split as an areal trait in Lingnan languages, mainly in the Tai and Sinitic languages spoken in this area. It uses the tone-box theory (which is hardly a theory) to illustrate the tonal split patterns and showcases abundant data in both groups of languages. The data is abundant and the topic is very interesting, but the materials are presented in a messy way and the organization of the paper has much space to improve. The paper requires extensive editing and revising in grammar and style.
### Major comments
The tone-box theory is used as the most important tool in the paper, but it can hardly be called a theory, rather a useful tool to illustrate the tonal split and tone merger patterns in the Tai and Sinitic languages. The tone-box chart is designed to capture all possible tonal splits according to the laryngeal features of initial consonants. The author's version differs from Gedner's only in the distinction between aspirated obstruents and voiceless sonorants in the upper tone series. It is better to concentrate on the most critical innovation and also to arrange the materials supporting this distinction.
The tone-box model is good in that it includes all possible groups of initial consonants which may cause a secondary tonal split, but the table is extremely complex. A better way of presenting the pattern is perhaps to only include the tonal categories (without example words) and to merge the cells which have the same tonal category (indicate the category using labels as A1A). Moreover, using different colors to represent tone categories makes it quite hard to understand how many tones are there in a language and it can be hard for someone who has no experience/difficulty seeing such subtle differences in color.
The arrangement of the paper needs some reorganization. An overview of the Tai languages, and also the languages in the Lingnan area should be given earlier in the paper. Different dialects/languages of the Tai languages are mentioned in different places and the reader has no big picture of the Tai languages. It is not clear why there are Sections 3.2 and 3.3 in the paper as the paper aims to investigate secondary tonal split.
First, section 3.2 is problematic. 濁上歸去 is not a tonal split, rather a tone merger. Even if the literary vs. colloquial readings are taken into consideration, it only shows there is some influence due to language contact. Zhuoshang gui Qu is simply the merger of the rising tone syllables with voiced initials to the departing tone. It is not tonal split at all. It is not conditioned by any laryngeal feature (of course by voicing, but voicing is the condition of primary tonal split). This section should be revised extensively or simply removed, because 濁上歸去 is not tonal split at all and is off point in the paper.
Section 3.3 is also puzzling in the current framework of the paper. Sonorant initials with a high register tone is interesting, but why put it in this paper, whose primary goal is to look at secondary tonal split? It can be removed with no harm and the contents can be put into another paper devoting to the issue of sonorant initials with a high register tone in Cantonese and other Chinese dialects.
The manuscript needs considerable revising and editing in English usage. It is quite hard to read and understand in its present form. I have made some comments and revisions in the pdf file, but a thorough revision is needed. Long sentences should be avoided as much as possible. I have also highlighted some sentences which are either ungrammatical or can be split up into several short sentences.
Some notations are used without explaination and the reader has to guess what they stand for. For example, on Page 8 (and also Page 9), 1234 are used without explanation of what they stands for. I guess they stand for different groups of the voiceless initials, but it is better to say that explicitly.
Another example is "tonal split/splitting". Both "tonal splitting" and "tonal split" are used in the paper, but it is recommended to use only one term (tonal split is more common in the literature) consistently in the paper.

Author Response
Dear Sir/Madam,
Thank you for your careful review of my manuscript and your suggestions for improvement! I have made some revisions to the manuscript according to the advice from you and the other reviewers. Some of the key points are as follows.
1. The basis of tone-box design involves the theory of tonal splitting and the treatment of various irregularities of tonal correspondence, and it certainly qualifies as a theory. For reasons of space, I cannot go into too much detail about the theories of tone-boxes that are not directly related to the focus of this paper; these can be seen in Gedney (1989 [1972]), Liao (2022), and some other studies.
However, I believe that the tone-box frameworks are not the only theory in this paper, but it is the improvement of the previously inadequate use of tone-box frameworks that is one of the theoretical contributions of this paper. I have followed another reviewer's suggestion, to revise the introduction (Section 1) to highlight the methodological aspects of this study more, focusing on how it improves an inadequate use of tone box theories in previous studies and maximizes the method, as well as how the current method applied can reveal new insights and cross-family correlations from the empirical data significantly increasing in the recent decades.
Moreover, I have moved the introduction to tonogenesis and tonal split, which was originally part of Section 1, to Section 2 as this part is relevant to the tone-box theory. 
2. I appreciate your suggestion that the Kra-Dai and Kam-Tai terms should be used consistently. However, since the former is the highest level of this family and the latter is a secondary branch, I have accepted your other suggestion to introduce Kra-Dai subgrouping in a figure (Figure 2) in the introduction. In addition, I have also checked the whole text to ensure the two terms Kra-Dai and Kam-Tai should be accurately used.
3. I appreciate your query about whether Sections 3.2 and 3.3 should be included in this paper because they are not tonal split patterns. I have changed “tonal split patterns” in the title of the manuscript to “tonal behavior” to avoid this seeming bug, but I have the following comments on this query.
For Section 3.2, for the criticism that lower register tonal splits, including zhuóshǎng guī qù, are tonal mergers rather than tonal splits, I think that zhuóshǎng guī qù is first a tonal split, i.e. it split from the original shǎng tone and then merges into the qù tone. This is a diachronic process. Moreover, not all Chinese dialects have zhuóshǎng guī qù, as in the Hong Kong Hakka dialect presented in this paper, where shǎng (B) tone conditioned by originally voiced obstruent has been merged to the upper píng tone (tone A1) rather than the qù tone. This means that it is a tonal split in the first place, and it is a later matter as to which tone it merges. In the previous studies of Kam-Tai tones, such a diachronic process of tonal split has been proven not to be a simple one-off. Therefore, I believe that this part as of secondary tonal split is not a problem.
However, Section 3.3 only discusses the association between upper tones and initial sonorants borrowed from Kam-Tai languages, which is not really part of the tonal split patterns. Nevertheless, because it is as deeply involved in the areal tonal behavior of Lingnan as the secondary tonal splits, I do think it is necessary to include this part in this paper. Therefore, I have revised the title of this manuscript.
4. I have accepted most of the other suggestions and minor comments on the other sections in your detailed proofreading. I also had the current revised version proofread by a native English speaker at your suggestion.
I have attached a copy of the revised manuscript. The first half shows the "track changes", such that any changes can be easily viewed by the editors and reviewers. The second half is the full version without “track changes” so that you can read it smoothly.
Thank you again for your review!
The Author
13 April 2023

Round 2
Reviewer 3 Report
The paper has imporved considerably after the first round of revision. The revision of the title to tonal behavior captures more precisely the central point of the paper and provides a wider scope for the study of the Sinitic and Kam-Tai languages in the Lingnan area. The paper would become better if the following questions are resolved.
Tonal behavior is better than tonal split in that it is more inclusive. But the term missed one very important point, that is the evolution of the tone, since tonal behavior can as well be a synchronic phenomenon. I think the authours would like to emphasize this point more, at least in the abstract.
I agree that 濁上歸去 involved tonal split first and then tone merger, but the tonal split is a primary split, rather than a secondary tone split, which is the tonal split within a register, as defined by the present paper. It is advisable that the authors should check the use of secondary tonal split when discussing 濁上歸去. The section heading on page 30 "lower register secondary tonal split" needs to be revised. Given the present heading, the reader would expect that different types of consonants conditioned the tonal split, but it is not the case. What the authors intended to say is that the merger of the earlier T3 syllables with voiced initial consonants merged into T4 with voiced initials, and this phenomenon is diffused from Chinese into Kam-Tai languages. This point is quite convincing, but it does not involve any secondary tonal split (as the manner of articulation of initials was not involved).
The use of some words needs to be consistent. For example, most places use tonal split, but there are some instances of tone split (page 11, 18, 46, etc)
The figures need to be made clearer and tidier for the final publication.
Author Response
Thanks for the suggestion to reinforce the evolution of the tone. In fact, the whole paper presents the diachronic development of tones, which is in fact a picture of the evolution of the tone. I have already added this point in the abstract.
I have a different view on the point that 濁上歸去 is not a secondary tonal split. Historically, voiced initials were not limited to voiced obstruents (全濁), but also includes voiced sonorants (次濁). In our conception of the primary tonal split in Chinese history, tone B (shang tone) was conditioned by both voiced obstruents and voiced sonorants, giving birth to yang shang or the lower Yang tone (B2); in contrast, yin shang was conditioned by voiceless initials. Such a pattern is well preserved in the colloquial readings of Lingnan Sinitic dialects such as Cantonese and Pinghua. Later, in the mainstream dialects of Chinese, the original yang shang tone split because of the condition of voiced obstruents (全濁), this produced a secondary tonal split that merged into one of the other tones, of which the dialects represented by Mandarin merged it into the C2 or lower qu tone (in contrast, the lower shang tone determined by the voiced sonorant did not merge into other tones, so it is regarded as the original lower register shang tone). In literary readings in Cantonese and Pinghua, it was the influence of Mandarin, which gave rise to the phenomenon of 濁上歸去 in these dialects. That is, the primary tonal split pattern (in colloquial readings representing) and the secondary split pattern (in literary readings) co-exist in Cantonese and Pinghua. Thus, in our scheme, 濁上歸去is a special type of secondary tonal split.
On pages 11, 18, and 46, "tone split" has been changed to "tonal split" as suggested, for keeping consistency.